# *Are all Frames Equal?* Active Sparse Labeling for Video Action Detection

**Aayush J Rana**
aayushjr@knights.ucf.edu

**Yogesh S Rawat**
yogesh@crcv.ucf.edu

Center for Research in Computer Vision (CRCV)
University of Central Florida

## Abstract

Video action detection requires annotations at every frame, which drastically increases the labeling cost. In this work, we focus on efficient labeling of videos for action detection to minimize this cost. We propose *active sparse labeling (ASL)*, a novel active learning strategy for video action detection. Sparse labeling will reduce the annotation cost but poses two main challenges; 1) how to estimate the utility of annotating a *single frame* for action detection as detection is performed at video level?, and 2) how these *sparse labels* can be used for action detection which require annotations on all the frames? This work attempts to address these challenges within a simple active learning framework. For the first challenge, we propose a novel frame-level *scoring mechanism* aimed at selecting most informative frames in a video. Next, we introduce a novel *loss formulation* which enables training of action detection model with these sparsely selected frames. We evaluate the proposed approach on two different action detection benchmark datasets, UCF-101-24 and J-HMDB-21, and observed that active sparse labeling can be very effective in saving annotation costs. We demonstrate that the proposed approach performs better than random selection, outperforming all other baselines, with performance comparable to supervised approach using merely 10% annotations. Project details available at https://sites.google.com/view/activesparselabeling/home

## 1 Introduction

Video action detection is a challenging problem with lot of applications in security [1, 2], autonomous driving [3, 4] and robotics [5, 6]. It requires spatio-temporal localization of an action in a video. This has led to innovative methods [7, 8, 9, 10, 11, 12] in recent years which rely on annotation on every frame of a video sample, which can be either bounding box [13] or pixel-wise [14]. This is different from action classification where a class label for each video is sufficient for training [15]. Therefore, it is challenging and costly to annotate an action detection dataset at a large-scale and existing datasets are much smaller in size [13, 14] as compared with classification datasets [15, 16, 17, 18, 19].

In this work, we focus on reducing the annotation effort for video action detection. The existing work in label efficient learning for action detection is mostly focused on semi-supervised [20, 21] or weakly-supervised approaches [22, 23, 24, 25]. These methods rely on either video-level annotations [25, 24], point annotations [21], pseudo-labels [20], or reduced bounding-box annotations [26] to reduce labeling effort. They rely on separate (often external) actor detectors and tube linking methods coupled with weakly-supervised multiple instance learning or pseudo-annotations, limiting the practical simplicity for general use. The video-level and pseudo-label approaches trade performance for saving annotations while the point and reduced bounding-box approach still have to annotate each instance to improve the performance. We argue that a lack of selection criteria for annotating only informative data is one of the limitations in these methods. Motivated by this, we propose *active*

36th Conference on Neural Information Processing Systems (NeurIPS 2022).

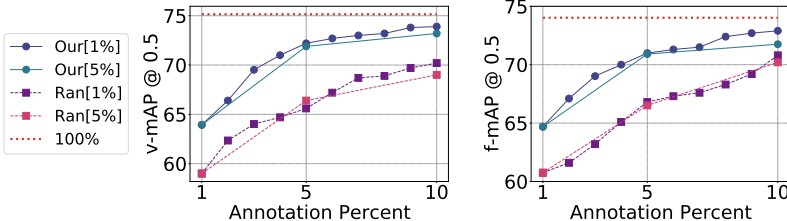

Figure 1: Comparison of our method with different annotation percent and step size against random selection method and fully-supervised method (100% annotations) on UCF-101. Results for 1% and 5% increment at 1%, 5%, and 10% annotations are shown for our and random selection [Ran].

*sparse labeling (ASL)* which bridges this gap between high performance and low annotation cost. *ASL* performs partial instance annotation (*sparse labeling*) by frame level selection where the goal is to annotate most informative frames, which are expected to be useful for activity detection task.

Sparse labeling will address the issue of high annotation cost, but there will be some challenges. It will require a frame level cost estimation which can determine the utility of each frame in a video. This estimation should be based on the frame's impact on action detection. Also, the traditional action detection methods require annotations on all the frames [8, 9, 10], therefore existing objective functions will not be effective with sparse labels. We propose a novel uncertainty based frame scoring mechanism for videos, termed as *Adaptive Proximity-aware Uncertainty (APU)*. APU estimates the frame's utility using the uncertainty of the detections and its proximity from existing annotations, determining *diverse* set of *informative* frames in a video which are more effective for learning the task of action detection. In addition, we propose a simple yet effective loss formulation, *Max-Gaussian Weighted Loss (MGW-Loss)*, which uses weighted pseudo-labeling for effective learning from *sparse labels*. Together, the proposed cost estimation algorithm based on APU and the MGW-Loss function helps in reducing the annotation cost while improving model performance at the same time.

We make the following contributions; 1) We propose **active sparse labeling (ASL)**, a novel active learning (AL) strategy for action detection where each instance is partially annotated to reduce the labeling cost. This is the **first work** focused on AL for video action detection to best of our knowledge, 2) We propose a **novel scoring mechanism** for selecting *informative* and *diverse* set of frames, 3) We also propose a **novel training objective** which helps in effectively learning from *sparse labels*.

We demonstrate the effectiveness of the proposed approach in optimizing annotation cost for video action detection in two different datasets, UCF-101-24 and J-HMDB-21. We reduce the annotation cost by $\sim$ **90%** with marginal drop in performance (Figure 1). We also evaluate the proposed approach for video object segmentation and demonstrate its *generalization capability*.

## 2 Related work

Recent works on action detection in videos uses a CNN based approach [27, 28, 8, 9, 29] to perform spatio-temporal localization of actors in videos. A common and effective theme is the two-stage approach using object detection methods [30, 28, 31, 32, 33] to detect actors per frame based on action classification models [15, 34, 35, 36, 37, 38, 39] and combine them into tubes using temporal aggregation [10, 8, 27, 11, 7] for classification. While some works need a separate region proposal network to detect potential actors [8, 40], using a complicated two-stage process, other recent works have proposed simplified single-stage approach [9, 41]. As our work is centered around AL for selective annotation, we prefer a single-stage approach for an efficient solution.

**Weak supervision:** Large datasets for action detection require costly annotations, therefore innovative approaches have been proposed for this task with weak and semi-supervised methods. Some of these works have been able to reduce annotation cost significantly for action detection [20, 24, 25, 42] by using only the video-level annotation. Other methods have proposed the use of dense point-level pseudo-annotations [21] or only few bounding-box per instance [23] with the same objective. A common drawback of these methods is the inferior performance compared to fully supervised methods, which limits their practical utility.

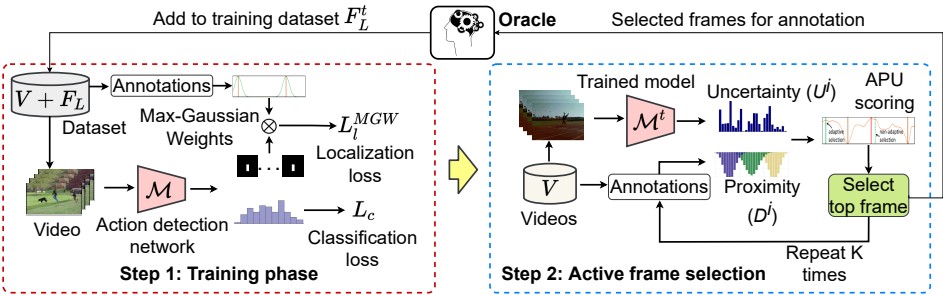

Figure 2: Overview of proposed approach. It consists of training and selection. During training the network is trained using existing annotations from the training set using *MGW-loss* to handle the sparse annotations. During iterative *APU* selection phase, the trained network is used to predict localizations on each frame of videos in the training set. Using these predictions, *APU* computes a score for each frame in a video to rank them and top *K* frames are sent to oracle for annotation.

**Active learning:** Active learning has been used to iteratively select unlabeled data for assigning labels based on certain utility factors [43, 44, 45, 46, 47]. Labeling a large set of data can often prove to be expensive and unnecessary, which is why AL can be vital in selecting related unlabeled data for further annotation in an iterative fashion. AL algorithms use uncertainty [46, 48, 49, 50, 51, 52], entropy [53, 54], heuristics and mutual information [45, 55, 56], core-set selection [57, 58] to select samples which are most likely to provide maximum utility to the learning algorithm. AL based classification algorithms are effective for different modalities such as images [45, 46, 44], videos [59, 60, 61], text [62, 63, 64] and speech [65]. Classification only needs class labels for an entire sample, making the scoring easier for the algorithm. However, extending that to a complex task such as object detection is challenging as it requires dense annotations in each sample [53, 66, 67, 68]. Extending that to videos adds extra level of complexities as it requires spatio-temporal annotations and selecting parts of video for extra annotation via AL algorithm is challenging. [69] performs frame selection using AL for object segmentation but does not leverage temporal aspect of video for avoiding sequential annotation, increasing overall annotation cost. There are *no existing methods* which focus on the problem of *active sparse labeling* in videos for *spatio-temporal detection task* and existing deep AL approaches are not applicable directly for this task.

## 3 Proposed method

We aim at reducing the annotation cost for labeling a set of videos $\mathcal{V} = \{V_1, ...V_N\}$ with $N$ videos to learn an action detection model $\mathcal{M}$. We start with an initial set of sparse labels $\mathcal{S}_L^0 = \{V_{cls}, F_L^0\}$ that consists of annotated frames with class label $V_{cls}$, where only a small number of frames $F_L^0$ are annotated. This initial set of sparsely annotated videos is used to initialize an action detection model $\mathcal{M}^0$. This initialized model $\mathcal{M}^0$ is then used to estimate a utility score for all the unlabeled frames $F_U^0$ from the set of videos $\mathcal{V}$. The goal is to automatically select frames from unlabeled set to be manually labeled and obtain new set of sparse labels $\mathcal{S}_S^0$ which is merged with $\mathcal{S}_L^0$ for a new labeled set $\mathcal{S}_L^1$. The number of additional frames are selected based on a total budget $B$ and they are annotated by an oracle (e.g. human annotator). The action detection model $\mathcal{M}$ is retrained using the new annotation set $\mathcal{S}_L^1$ and an updated model $\mathcal{M}^1$ is obtained. This process is repeated until we find a set $\mathcal{S}_L^F$ with several annotated frames in the videos $\mathcal{V}$ such that $\mathcal{M}^\mathcal{F}$ meets the target performance or the total budget $B$ is exhausted. An overview of the proposed approach is shown in figure 2.

### 3.1 Active sparse labeling

**Sparse labeling:** We hypothesize that some frames will have more utility than others for learning action detection due to several factors, such as lack of motion, variation in action dynamics, redundancy in appearance or redundancy in action. In sparse labeling, we annotate only $l$ frames $f_{v,l}$ in a video $v$ instead of labelling all of them, leaving a set of $u$ unannotated frames $f_{v,u}$. Therefore, it avoids annotation of frames with lower utility and helps in reducing the overall labeling cost. Each video $v$ has a class label, denoted as $v_{cls}$, for the action category and a set of $l$ annotated frames $f_{v,l}$ which indicates the localization of actions.

**Uncertainty as frame utility:** In each AL cycle, our goal is to select video frames for labeling which will have the highest utility for learning action detection. Uncertainty provides a measure to estimate model's confidence on its decision and has been used for selecting informative samples in existing works [70, 71, 72, 61]. These works are focused on classification and therefore the uncertainty is computed for the entire sample. Instead, we require informativeness of each frame in a video which is different from these works as it is computed for partial sample. The action detection model $\mathcal{M}$ provides spatio-temporal localization for the entire video and we propose to use the pixel-wise confidence score of localization on each frame to estimate frame-level uncertainty. We use MC-dropout [73] to estimate the model's uncertainty for each pixel in the video. MC-dropout is a more efficient form of uncertainty estimation compared to using a Bayesian neural network and is easier to implement [73, 74]. The uncertainty is estimated over $T$ different trials and this score is averaged over all the pixels in a frame. For a given video $v$ with $I$ frames, the uncertainty $U^i$ for the $i^{th}$ frame over $T$ trials is computed as,

$$U^{i \in [1, I]} = \frac{1}{I^p} \sum_{h=1}^{I^p} \frac{1}{T} \sum_{j=1}^{T} -log(P(v_h^i, j)) \tag{1}$$

where $I^p$ is the total number of pixels in a frame, and $P(v_h^i, j)$ represents the model prediction for $h^{th}$ pixel in the $i^{th}$ frame of video $v$ during the $j^{th}$ trial.

**Adaptive proximity-aware uncertainty:** Unlike images, the motion in videos has some continuity and it is highly likely that the frames close to each other will have similar uncertainty scores. Therefore selecting frames merely based on uncertainty will favor adjacent frames which may have similar utility for learning action detection. To overcome this issue, we propose a selection mechanism, termed as *Adaptive Proximity-aware Uncertainty (APU)*, which ensures that the selected frames have diversity in the temporal domain. APU scoring incorporates a distance measure into cost estimation and uses their *proximity* to the existing annotated frames. As we select more frames, this distance measure should *adapt* to the additional selected frames. We use a normal distribution $\mathcal{N}(\mu, \sigma^2)$ for distance measure $D$, where each annotated frame has its own distribution centered around its temporal location in the video. Given a video with $K$ annotated frames, the distance measure $D^i$ for the $i^{th}$ frame of the video is computed as,

$$D^i = 1 - \sum_{j=1}^{K} \varphi_i^j e^{-\frac{1}{2}(\frac{i - \mu_j}{\sigma})^2}. \tag{2}$$

where $D^i$ is distance measure for unannotated frame $i$ from annotated frame $j$, the distribution $\mathcal{N}$ for $j^{th}$ annotated frame is centered at frame $j$ with $\mu_j$ mean and $\sigma$ variance, and $\varphi_i^j \in [0, 1]$ is the mask to select the closest distribution for $i^{th}$ frame. The value of mask $\varphi_i^j$ will be 1 for $j^{th}$ distribution if it is closest to the $i^{th}$ frame, otherwise it will be 0. APU scoring uses both uncertainty and proximity and therefore prefers frames with high uncertainty and ensures temporal diversity. The overall APU score $\mathcal{U}_{APU}^i$ for a given frame is computed as,

$$\mathcal{U}_{APU}^i = \lambda U^i + (1 - \lambda) D^i \tag{3}$$

where $\lambda$ is used to control the contribution from uncertainty and temporal diversity. It is set to 0.5 for equal contribution in our formulation where $U, D$ are both normalized in range (0,1).

**Informative frame selection:** Once we get $\mathcal{U}_{APU}$ score for all the frames in $\mathcal{V}$ videos, we select the frame with highest score globally and then score the remaining frames again with the adapted distance measure. The re-scoring is necessary to reduce probability of picking frames around same region, since a region doing poorly is likely to have more frames which scored higher in the selection process. We only need to recompute the distance measure, which is computationally inexpensive. The entire selection algorithm is provided Appendix. Once we have $F_{annot}$ frames selected as per our budget, they are annotated by an oracle and the training set is updated with these new annotations. This completes one AL cycle and the model $\mathcal{M}$ is trained using the updated annotations.

**Non-activity suppression:** If all pixels in a frame are considered to compute its utility, non-activity regions may negatively influence the score as the model easily determines background pixels compared to the actual action region in a frame. A low uncertainty score from background pixels

will lower the overall frame uncertainty even if the activity region has high uncertainty, especially in videos with a relatively larger background area as compared to the actual action region. Therefore, we ignore pixels which are predicted as background (true negatives and false negatives) with high confidence (using a threshold $\tau$) when computing the frame-level uncertainty. This might exclude some foreground pixels (false negatives) from the uncertainty estimation. However, these pixels will not be useful even if we use them as they are predicted as background due to low uncertainty.

## 3.2 Learning from sparse labels

Given a video clip $V = \{f_1, f_2, ... f_N\}$ with $N$ frames where $K$ frames are annotated such that $K < N$, we have to detect the action through the entire clip. A traditional action detection network is trained with the help of two different objectives, a classification loss $L_c$ for action category and a localization loss $L_l$ for spatio-temporal detection [8]. The classification loss $L_c$ is computed for the entire video clip and the localization loss is computed for every frame in the video.

Sparse labeling will not allow us to compute the localization loss $L_l$ on every frame due to missing annotations. The localization loss $L_l$ with sparse labeling can be computed as, $L_l = \sum_{i=1}^{N} \beta^i L_l^i$. Here, $L_l^i$ represents localization loss in the $i^{th}$ frame and $\beta^i \in [0, 1]$ indicates masking, which will be 1 for annotated frames and 0 otherwise. The masking strategy only uses the annotated frames for learning, therefore it is not quite effective. In a contrastive approach, we can use all the frames for learning by generating their *pseudo-labels* with the help of interpolation of annotations from neighboring frames. This will allow us to use all the frames but incurs noise from the pseudo-labels.

**Max-gaussian weighted loss:** We propose a simple loss formulation which benefits from both, masking and pseudo-labels. We hypothesize that the pseudo-labels close to ground-truth labels will be more reliable. Based on this, we propose *Max-Gaussian Weighted Loss (MGW-Loss)* which discounts the approximated pseudo-labels as they will not be as reliable as the actual ground-truth. We compute localization loss for each frame using both available and pseudo-labels, where the pseudo-labels have a varying weight in the overall loss component. The approximated annotations will not have a similar weight as their distance from the annotated frames will vary. We use a mixture of Gaussian distribution $w \in \{1..W\} \sim \mathcal{N}_w(\mu_{gt}, \sigma^2)$ to assign the weight to each frame, given $gt \in \{1..K\}$ actual ground-truth frame location as the mean of the distribution and $\sigma$ is the variance of the distribution. We define the weighted localization loss $L_l^{MGW}$ as,

$$L_l^{MGW} = \sum_{i=1}^{N} (\sum_{j=1}^{K} \phi_j^i e^{-\frac{1}{2}(\frac{i-\mu_j}{\sigma})^2}) L_l^i. \tag{4}$$

Here $L_l^i$ is the localization loss of $i^{th}$ frame for any video, $\mu_j$ is the frame location for $j^{th}$ annotated frame, and $\phi_j^i \in [0, 1]$ is the mask to select the max distribution for $i^{th}$ frame. The value of mask $\phi_j^i$ will be 1 for $j^{th}$ distribution if it has the maximum probability among all Gaussians at location of $i^{th}$ frame, otherwise it will be 0. The value of $\sigma$ controls the weighting mechanism and it has two extremes. The high variance is equivalent to interpolation where all the frames will have equal weights and low variance is equivalent to masking where weights of pseudo-labels will be 0. Details on influence of $\sigma$ provided in appendix C.1.

## 3.3 Action detection model

Video action detection is a challenging problem and the existing methods usually follow a complex pipeline [8, 10, 75]. Region proposal based approach has been found to be exceedingly effective [40], which has also been extended to tube proposals [8, 10]. However, training these two-step methods is not efficient, especially when we have to develop an iterative framework for AL. We follow a simpler approach where classification and detection can be done in an end-to-end training [9]. We simplified VideoCapsuleNet [9] further and replaced the 3D routing with 2D routing [76] which makes it more efficient in terms of memory and training speed. We then added dropout layers for uncertainty and used *MGW-Loss* from Eq. 4 to handle sparse labels. To handle sparse labels, we get the frame-wise weight from max-Gaussian weighted method and adjust the loss using this weight. Following [9], the network is trained using margin-loss for classification and binary-cross entropy loss for spatio-temporal localization.

Table 1: Comparison between different baseline methods in UCF-101 and J-HMDB dataset for different frame annotation percent. * is extended to video action detection using same backbone detector network as ours. [G: Gal et al.[73], A: Aghdam et al.[53]]

| | UCF-101 | | | | | | J-HMDB | | | | | |
| | f-mAP@0.5 | | | v-mAP@0.5 | | | f-mAP@0.5 | | | v-mAP@0.5 | | |
| Method | 1% | 5% | 10% | 1% | 5% | 10% | 3% | 6% | 9% | 3% | 6% | 9% |
|---|---|---|---|---|---|---|---|---|---|---|---|---|
| Random | 60.7 | 66.5 | 69.3 | 59.2 | 66.4 | 69.9 | 58.3 | 69.3 | 71.6 | 57.4 | 64.6 | 70.4 |
| Equidistant | 61.8 | 66.2 | 68.4 | 61.7 | 67.2 | 69.0 | 57.4 | 67.5 | 71.4 | 56.9 | 64.9 | 66.8 |
| G* [73] | 60.9 | 66.7 | 68.9 | 59.4 | 66.8 | 69.1 | 58.2 | 66.7 | 67.5 | 57.4 | 66.8 | 67.4 |
| A* [53] | 61.4 | 67.9 | 69.8 | 60.1 | 67.9 | 70.0 | 58.8 | 71.2 | 71.1 | 57.7 | 66.7 | 71.2 |
| **Our** | **64.7** | **70.9** | **71.7** | **63.9** | **71.8** | **73.2** | **68.8** | **74.1** | **74.5** | **65.6** | **70.8** | **74.0** |

## 4 Experiments

**Datasets and evaluation metrics:** We evaluate our approach on three different datasets, **UCF-101** [13], **J-HMDB** [14] and **YouTube-VOS** [77]. UCF-101 has 3207 videos from 24 different classes with spatio-temporal bounding box annotations. J-HMDB dataset contains 928 videos from 21 classes with pixel-level spatio-temporal annotations. AVA [78] is a large-scale dataset with focus on action detection on single keyframe. This makes AVA not suitable for sparse frame selection as it does not provide spatio-temporal annotations. Similarly, MAMA [79] is a challenging dataset with low baseline detection scores (<1%), limiting sparse annotation study. We further evaluate our method on **YouTube-VOS** [77] for video object segmentation to demonstrate its generalization capability. YouTube-VOS consists of 3471 training videos (65 categories) with pixel-level annotation for multi-object segmentation. Following prior action detection works [80, 22] on UCF-101 and J-HMDB datasets, we compute the spatial IoU for each frame per class to get the frame average precision score and compute the spatio-temporal IoU per video per class to get the video average precision score score. This is then averaged to obtain the f-mAP and v-mAP scores over various thresholds. For video segmentation, we evaluate the avg. IoU ($\mathcal{J}score$) and the avg. boundary similarity ($\mathcal{F}score$) [77].

**Implementation details:** We implement our method in PyTorch [81]. In video action detection architecture, we use I3D encoder head [15] with pre-trained weights from the Charades dataset [82]. We use Adam optimizer [83] with a batch size of 8 and train for 22K iterations in each active learning cycle (details in appendix G.3). We use dropout for generating uncertainty similar to [73] by enabling it during inference. For YouTube-VOS task, we use two existing methods [77, 84]. We use $\tau = 0.9$ for non-active suppression and $\sigma = 1.3$ for Eq. 2 and Eq. 4, which were empirically determined (details in appendix C).

**Sparse learning settings:** In the initialization stage, we assume the availability of annotations for $I\%$ of frames in each video in $\mathcal{V}$ to make sparse annotation set $\mathcal{S}_L^0$. These frames are randomly selected for the first stage. We use 1%, 3%, and 5% initial frames for UCF-101, J-HMDB, and Youtube-VOS respectively, determined empirically based on each dataset size. We assign annotation cost for each frame as $C_{frame} = Actor \times Clicks$ based on clicks per actor (bounding box/pixels).

### 4.1 Baseline methods

We explore several baselines to understand their limitations on video action detection. First, we use random and equidistant frame selection where random selection select the frames at random in each stage, equidistant uses equal distance between the frames during selection. Next, we extend existing AL methods used in image-based object detection [53, 73] to video action detection, where we score each frame using their algorithm for frame selection. We improve upon the uncertainty sampling for video level selection from [61] and compute uncertainty at pixel-level in all our baselines. We train all baselines using same action detection backbone for a fair comparison. We have random, equidistant, uncertainty-based [73] and entropy-based [53] approaches as baseline methods to compare against.

### 4.2 Results

**Analysis of baseline methods:** We evaluate random, equidistant, entropy-based [53] and uncertainty-based [73] selection methods as baselines and compare with our approach in Table 1. While all baselines are effective for AL in image-based detection/classification tasks, we demonstrate that for video action detection prior methods [53, 73] perform similar or worse than random

Table 2: Evaluation of the proposed method on **UCF-101** and **J-HMDB**.

| | UCF-101 | | | | | J-HMDB | | | |
| Annot | v-mAP@ | | f-mAP@ | | Annot | v-mAP@ | | f-mAP@ | |
| Percent | 0.3 | 0.5 | 0.3 | 0.5 | Percent | 0.3 | 0.5 | 0.3 | 0.5 |
|---|---|---|---|---|---|---|---|---|---|
| 1% | 89.01 | 63.94 | 83.85 | 64.69 | 3% | 95.15 | 65.56 | 89.94 | 68.78 |
| 5% | 90.95 | 71.89 | 88.71 | 70.91 | 6% | 95.20 | 70.75 | 93.09 | 74.09 |
| 10% | 91.12 | 73.20 | 88.72 | 71.75 | 9% | 95.58 | 74.01 | 92.67 | 74.50 |
| 100% | 91.49 | 75.15 | 89.08 | 74.02 | 100% | 96.39 | 75.75 | 93.74 | 74.91 |

Table 3: Comparison with state-of-the-art methods. We evaluate our approach using v-mAP and f-mAP scores using only 10% annotations. 'Video' uses video-level class annotations and 'Partial' uses sparse temporal and spatial annotations. V: video labels, P: points, B: bounding box, O: off-the-shelf detector. **f@** denotes **f-mAP@**

| Method | Annot Percent | V | P | B | O | UCF-101 f@ 0.5 | v-mAP@ 0.1 | 0.2 | 0.3 | 0.5 | J-HMDB f@ 0.5 | v-mAP@ 0.1 | 0.2 | 0.3 | 0.5 |
|---|---|---|---|---|---|---|---|---|---|---|---|---|---|---|---|
| *Fully supervised* | | | | | | | | | | | | | | | |
| Peng et al. [7] | 100% | | | | | 65.7 | 77.3 | 72.9 | 65.7 | 35.9 | 58.5 | - | 74.3 | - | 73.1 |
| TCNN [8] | 100% | | | | | 67.3 | 77.9 | 73.1 | 69.4 | - | 61.3 | - | 78.4 | - | - |
| Gu et al. [78] | 100% | | | | | 76.3 | - | - | - | 59.9 | 73.3 | - | - | - | - |
| ACT [85] | 100% | | | | | 69.5 | - | 76.5 | - | - | - | - | 74.2 | - | 73.7 |
| STEP [10] | 100% | | | | | 75.0 | 83.1 | 76.6 | - | - | - | - | - | - | - |
| VidsCapsNet [9] | 100% | | | | | 78.6 | 98.6 | 97.1 | 93.7 | 80.3 | 64.6 | 98.4 | 95.1 | 89.1 | 61.9 |
| *Weakly/Semi-supervised* | | | | | | | | | | | | | | | |
| Mettes et al. [20] | Video | ✓ | | | ✓ | - | - | 37.4 | - | - | - | - | - | - | - |
| Escorcia et al. [24] | Video | ✓ | | | | - | - | 45.5 | - | - | - | - | - | - | - |
| Zhang et al. [25] | Video | ✓ | | | ✓ | 30.4 | 62.1 | 45.5 | - | 17.3 | 65.9 | 81.5 | 77.3 | - | 50.8 |
| Arnab et al. [42] | Video | ✓ | | | ✓ | - | - | 61.7 | - | 35.0 | - | - | - | - | - |
| Weinz. et al. [26] | Partial | ✓ | | ✓ | ✓ | 63.8 | - | 57.3 | - | 46.9 | 56.5 | - | - | - | 64.0 |
| Mettes et al. [21] | Partial | ✓ | ✓ | | | - | - | 41.8 | - | - | - | - | - | - | - |
| Cheron et al. [23] | Partial | ✓ | | | ✓ | - | - | 70.6 | - | 38.6 | - | - | - | - | - |
| Kumar et al. [86] | 20% | ✓ | ✓ | | | 69.9 | - | 95.7 | - | 72.1 | 64.4 | - | 95.4 | - | 63.5 |
| Ours | 10% | ✓ | ✓ | | | 71.7 | 98.1 | 96.5 | 91.1 | 73.2 | 74.5 | 99.2 | 98.4 | 95.6 | 74.0 |
| Ours | 100% | | | | | 74.0 | 98.3 | 96.9 | 91.5 | 75.2 | 74.9 | 99.2 | 99.2 | 96.4 | 75.8 |

or equidistant methods. The lack of temporal information prohibits prior methods to select frames effectively as videos have sequential frames in same region with high uncertainty. Our approach accounts for the temporal continuity and outperforms all baselines including prior AL based methods [73, 53] consistently on both dataset for all annotation percent. This demonstrates that extending image-based methods are not well suited for video action detection task as shown in Figure 3.

**Evaluation of proposed method:** We evaluate our approach on UCF-101 and J-HMDB for action detection and compare with fully-supervised training in Table 2. For **UCF-101** we initialize with 1% of labelled frames and train the action detection model with a step size of 5% in each cycle. We achieve results very close to full annotations (v-mAP@0.5, 73.20 vs 75.12) using only 10% of annotated frames, which is a huge reduction (90%) in the annotation cost. For **J-HMDB**, we initialize with 3% labels as it is a relatively smaller dataset and it is challenging to train an initial model with just 1% labels. Here, we obtain results comparable with 100% annotations with only 9% of labels.

**Comparison to prior weakly/semi-supervised approach:** We compare to prior weakly/semi-supervised action detection approach [23, 24, 21, 20, 26, 86] in Table 3 and explain their limitations. [26] uses external human and instance detectors to build tubes aligned with 1-5 random spatially annotated GT frames per tube. This incurs larger annotation cost without any frame selection metric while having low performance. [20, 21, 42] follow Multi Instance Learning (MIL) approach, where [20] uses off-the-shelf actor detectors to generate pseudo-annotations and [21] relies on user input for point annotation *for every frame*, requiring large annotation cost. [42] expands on MIL approach combined with tubelets generated by an off-the-shelf human detector. While MIL based approach requires less oversight, it also suffers from reduced performance, even with state-of-the-art detectors.

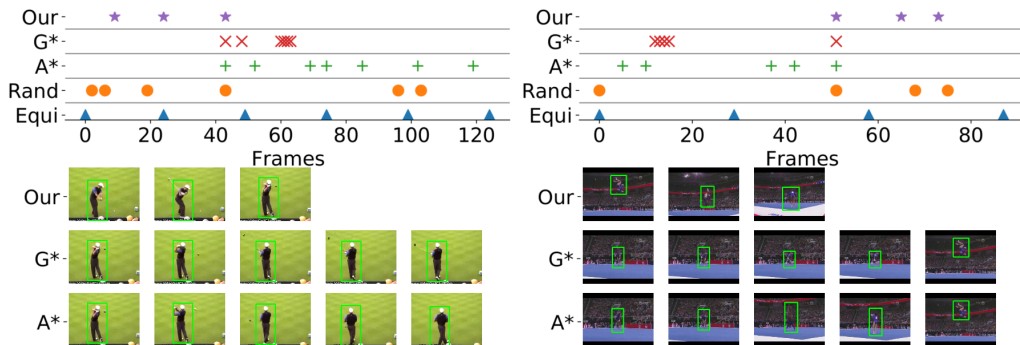

Figure 3: Analysis of frame selection using different methods. The x-axis represents all frames of the video, with each row representing a baseline method. The markers for each method mark the frames selected using that method. For both samples, our method selects distributed frames centered around action region, Gal et al [73] **[G\*]** selects frame around same region since there is no distance measure and Aghdam et al. [53] **[A\*]** selects slightly more distributed frames but those are not from crucial action region. [G\*:Gal et al[73], A\*:Aghdam et al[53], Rand: Random, Equi: Equidistant]

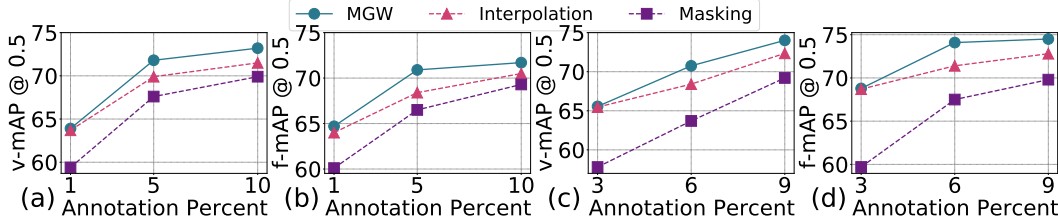

Figure 4: Comparison of different loss functions for UCF-101 (a-b) and J-HMDB (c-d).

[24] uses actor detector with video-level label to perform action detection, using a less involved approach as [42], but both have high label noise and low performance. [86] uses consistency regularization to train with unlabeled data in semi-supervised fashion. [23] uses discriminative clustering instead of MIL to assign tubelets to action label with various level of supervision, [25] uses combination of different actor detectors to build tube to train with video labels. They rely on multiple off-the-shelf components to generate the tubelets and suffer from low performance. [25] and [26] report their J-HMDB results using bounding-box annotation instead of the fine-grained pixel-wise annotation due to their design limitation to use external bounding-box detector for tube generation. Our approach does not rely on such detectors and can work with both bounding-box (UCF-101) and pixel-wise (J-HMDB) annotation and is comparable to the supervised performance.

### 4.3  Ablations

**Effect of loss function:**   We evaluate the effectiveness of *MGW-Loss* for video action detection with sparse labels and compare it with baseline masking and interpolation based loss in figure 4. The proposed *MGW-Loss* learns better in sparse label conditions due to the approximated ground truth frames from interpolation. Without the approximated frames, the formulation in Eq. 4 will reduce to masking loss as $\sigma \rightarrow 0$. Masking computes loss only on the sparse ground truth and does not perform as well as the *MGW-loss* with interpolated ground truth as seen in figure 4. Our Gaussian based interpolation adapts better for approximated labels compared with simple interpolation due to having different weight for each frame based on their distance from real ground truth annotation.

**Effect of selection criteria:**   We compare how commonly used entropy and uncertainty-based selection methods perform against proposed *APU* algorithm when using the same loss formulation from Eq. 4. Figure 5 shows that *APU* has optimum frame selection as it encourages diverse selection by using adaptive distance to existing frames for the scoring process. Following [53], entropy based selection has a less effective fixed distance filter to avoid nearby frames. The uncertainty method

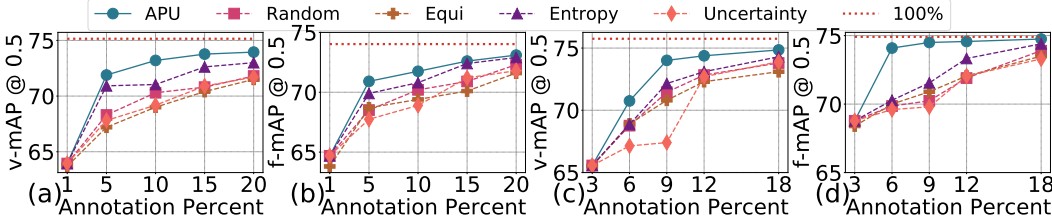

Figure 5: Comparison of different frame selection methods combined with MGW-Loss showing v-mAP and f-mAP scores for IoU @ 0.5 for (a-b) UCF-101 evaluation up to 20% (∼40k frames) data annotation. (c-d) J-HMDB evaluation up to 18% (∼3800 frames). Our *APU* approach gets better performance at a lower annotation percentage (lower annotation cost).

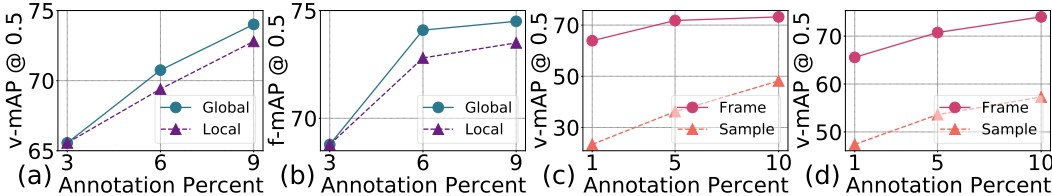

Figure 6: Analysis on sample selection strategy. (a-b) Global vs local frame selection strategy using *APU* on J-HMDB. (c-d) Frame vs sample selection for UCF-101 (c) and J-HMDB(d).

lacks any distance component and performs worse than random or equidistant, selecting frames from nearby regions as seen in figure 3.

**Annotating more frames**   We also evaluate adding additional frames until the scores start to saturate, shown in figure 5. We see that for UCF-101 at 20% annotation (∼40k frames) all methods score close to each other. Similarly, J-HMDB dataset has similar convergence pattern at 18% annotation (∼3800 frames). This demonstrates that while the frame selection eventually converges with more data, our approach gets better score at an earlier stage, saving overall annotation cost.

### 4.4 Discussions

**Variation in budget steps:**   Lower budget steps enables selection of fewer frames with high utility in each step instead of selecting more frames with low utility in higher budget steps. As the annotation set is more curated in each step in lower steps, we end up with better frames for the same annotation budget as higher steps. We evaluate the effect of using step size of 1% and 5% in figure 1 for UCF-101 dataset, starting from 1% till 10%. Step size of 1% has constantly better v-mAP and f-mAP score throughout, showing that smaller steps give greater performance. However, smaller step size needs more iterations, taking more time as a trade-off for better performance.

**Local vs. global selection:**   The proposed approach is focused on sparse labeling where frames with high utility within a video are selected for annotations. However, it is important to note that videos as a whole have varying utility. To exploit this aspect, we explore two different frame selection strategies, local selection and global selection. In local selection, each video has a fixed budget $b/N^v$, where $b$ is budget per cycle and $N^v$ is the total number of videos in our training set. However, frames in global selection are taken from a global pool which includes frames from all videos, ranking based on overall dataset utility. Figure 6 (a-b) shows that global selection outperforms local selection strategy, emphasizing that some videos can be more informative than others as confirmed in figure 7.

**Sample vs. frame selection:**   We follow [21] and annotate the entire sample (video) instead of finding the most useful frames within each sample. We compute pixel level uncertainty which is averaged over all the pixels in a frame using Eq. 1 and then averaged over all frames in a video to get the video level score. While this approach is simpler, it has higher cost during annotation with lower data variation. Let us assume a fixed cost of $c$ per frame with $f$ frames to annotate, we can assume a budget of $B = c \times f$. We could distribute the frames across the set by picking only few

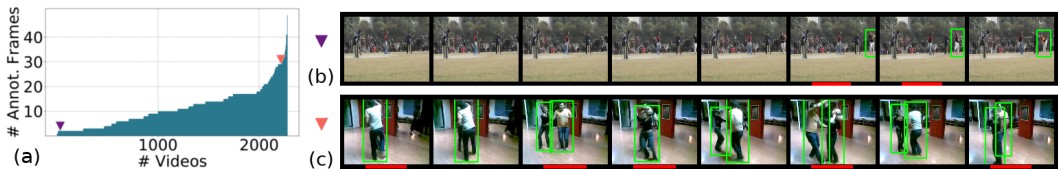

Figure 7: Sparse selection analysis. (a) Histogram showing number of frames selected per video using our method on UCF-101. Videos on the right show two samples from extreme ends of this histogram as marked in the plot. (b) Samples for cricket bowling class with APU selected frames on red marker (APU selects only two frames for 1 action instance). (c) Samples for salsa spin class (APU selects multiple frames (red) as each spin instance is visually diverse).

Table 4: Comparison of the proposed method on YouTube-VOS dataset with baseline AL methods using STCN [84]. $A$ = Aghdam et al. [53], $G$ = Gal et al. [73]. * is extended to video object segmentation using same network as ours.

| Method | Overall | | | $\mathcal{J}_S$ | | | $\mathcal{J}_U$ | | | $\mathcal{F}_S$ | | | $\mathcal{F}_U$ | | |
|---|---|---|---|---|---|---|---|---|---|---|---|---|---|---|---|
| | 10% | 20% | 30% | 10% | 20% | 30% | 10% | 20% | 30% | 10% | 20% | 30% | 10% | 20% | 30% |
| Random | 28.4 | 42.3 | 42.5 | 29.1 | 42.9 | 43.8 | 25.8 | 38.5 | 38.6 | 30.2 | 44.3 | 45.0 | 28.4 | 43.5 | 42.7 |
| $A$ * [53] | 30.1 | 45.6 | 47.2 | 31.5 | 45.4 | 47.6 | 26.7 | 43.4 | 47.9 | 22.8 | 46.7 | 48.8 | 17.6 | 46.8 | 44.6 |
| $G$ * [73] | 27.9 | 45.1 | 48.8 | 28.5 | 50.8 | 48.5 | 24.8 | 42.0 | 46.6 | 29.7 | 42.1 | 49.8 | 28.7 | 45.5 | 50.4 |
| **Our** | **31.7** | **58.6** | **66.7** | **33.6** | **58.2** | **66.7** | **27.8** | **54.3** | **61.5** | **35.2** | **60.6** | **69.1** | **30.1** | **60.9** | **69.7** |

important frames from each video, which would increase variation in the training set. However, if we annotate entire sample, there will be many redundant annotations with little gain, which is why frame selection performs better for action detection task as observed in figure 6 (c-d).

### 4.5 Generalization beyond action detection

We test generalization of proposed cost and loss function for video object segmentation task on the YouTube-VOS 2019. Table 4 shows that our proposed selection approach gets better $\mathcal{J}$ and $\mathcal{F}$ scores for video segmentation task compared to baseline AL methods and random frame selection method. We provide further details in appendix F and G.2.

### 4.6 Limitations

As most AL methods, this approach reduces annotation cost with multiple selection iterations, being more time consuming compared to weakly-supervised approaches. As we only measure frames within same video, future work can focus on evaluating each video's utility as well for the task.

## 5 Conclusion

In this work we present active sparse labeling (ASL), a novel approach for label-efficient video action detection. The proposed approach uses an uncertainty-based scoring mechanism for selecting informative and diverse set of frames for action detection. In addition, we also propose a simple yet effective loss formulation which can be used to train a model with sparse labels. The proposed approach is promising in saving annotation costs and we show that merely 10% of labels can achieve performance comparable to fully supervised methods. We further demonstrate the generalization capability of the proposed approach for video object segmentation.

## 6 Acknowledgements

This research is based upon work supported in part by the Office of the Director of National Intelligence (Intelligence Advanced Research Projects Activity) via 2022-21102100001 and in part by University of Central Florida seed funding. The views and conclusions contained herein are those of the authors and should not be interpreted as necessarily representing the official policies, either expressed or implied, of ODNI, IARPA, or the US Government. The US Government is authorized to reproduce and distribute reprints for governmental purposes notwithstanding any copyright annotation therein.

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
