# *Are all Frames Equal?* Active Sparse Labeling for Video Action Detection (Supplementary Material)

**Aayush J Rana**
aayushjr@knights.ucf.edu

**Yogesh S Rawat**
yogesh@crcv.ucf.edu

Center for Research in Computer Vision (CRCV)
University of Central Florida

## A    Appendix

We present additional analysis, experiments and details for various components in the main manuscript. We perform detailed ablations on different loss function and analysis of using varying $\sigma$ for the proposed *MGW-Loss*, which provides detailed explanation for effect of $\sigma$ on performance and validates our choice of $\sigma$ for all experiments in the main paper. We also explain the potential social impact our work can have (positive and negative) in detail.

## B    Potential Social Impact

This work can have positive and negative impact on future learning task related to complex video understanding by reducing the amount of annotations needed for training a model. On the positive side, while the iterative selection can take longer initially, it will save multiple hours on annotation for a large scale dataset. A negative impact of this approach is its use to select and annotate data for surveillance and intrusive monitoring. However, this can improve general video understanding at large since the cost of annotating large dataset will be significantly less with this approach.

## C    Loss function details

The proposed *MGW-Loss* formulation is able to handle sparse annotations via interpolation of known ground truth annotations and a Gaussian based weight estimation. Interpolation gives the in-between annotation frames for a clip with sparse ground truth annotation, which allows the use of traditional loss computation directly. However, since interpolation can be of low quality and not necessarily on the correct region of frame, using it as is affects the network's learning process due to adverse noise. While we could directly mask frames with ground truth annotation and only use those for loss computation, it is not able to train a model with low loss for initial frame selection algorithm. Table 1 and 2 shows the performance gap for model trained with different loss types (masking, interpolation and *MGW* loss). To utilize the ground truth annotated frames fully while giving only partial importance to the interpolated frames, we propose our *Max-Gaussian Weighted Loss* which uses a Gaussian distribution to give weight to the neighboring interpolated annotations. The Gaussian is fixed around each ground truth annotation and the value for each frame is used as its weight during loss computation. So an annotated frame further away from actual ground truth will have very low weight compared to one near to the actual ground truth. Figure 1 visually demonstrates the annotations for loss computation using masking, interpolation and proposed Gaussian interpolation

36th Conference on Neural Information Processing Systems (NeurIPS 2022).

Table 1: Comparison between different loss functions in UCF-101 for different percent of annotated frames per video. For each setting, we train the model using the masking loss, the interpolation loss and the proposed *MGW* loss with $\sigma = 1.3$.

| Annotation Percent | Loss Type | v-mAP@ | | | | f-mAP@ | | | |
|---|---|---|---|---|---|---|---|---|---|
| | | 0.1 | 0.2 | 0.3 | 0.5 | 0.1 | 0.2 | 0.3 | 0.5 |
| 1% | Masking | 97.89 | 93.45 | 86.53 | 59.4 | 91.04 | 88.25 | 83.54 | 60.10 |
| 1% | Interpolation | 97.99 | 94.24 | 87.69 | 63.71 | 91.14 | 88.45 | 83.69 | 64.04 |
| 1% | MGW | 98.07 | 95.37 | 89.01 | 63.94 | 91.28 | 88.41 | 83.85 | 64.69 |
| 5% | Masking | 98.01 | 95.10 | 89.98 | 67.60 | 93.18 | 90.07 | 87.78 | 66.50 |
| 5% | Interpolation | 98.05 | 95.30 | 90.28 | 69.41 | 93.21 | 91.30 | 88.16 | 68.41 |
| 5% | MGW | 98.12 | 96.01 | 90.95 | 71.89 | 93.38 | 91.28 | 88.71 | 70.91 |
| 10% | Masking | 98.1 | 95.84 | 90.23 | 69.90 | 94.02 | 92.05 | 88.30 | 69.30 |
| 10% | Interpolation | 98.18 | 96.43 | 91.12 | 71.52 | 94.07 | 92.07 | 88.10 | 70.57 |
| 10% | MGW | 98.14 | 96.46 | 91.12 | 73.20 | 94.05 | 92.03 | 88.72 | 71.75 |

Table 2: Comparison between different loss functions in JHMDB for different percent of annotated frames per video. For each setting, we train the model using the masking loss, the interpolation loss and the proposed *MGW* loss with $\sigma = 1.3$.

| Annotation Percent | Loss Type | v-mAP@ | | | | f-mAP@ | | | |
|---|---|---|---|---|---|---|---|---|---|
| | | 0.1 | 0.2 | 0.3 | 0.5 | 0.1 | 0.2 | 0.3 | 0.5 |
| 3% | Masking | 98.01 | 96.18 | 90.26 | 57.81 | 95.10 | 92.39 | 86.84 | 59.74 |
| 3% | Interpolation | 98.80 | 96.70 | 95.03 | 65.47 | 95.40 | 93.55 | 89.02 | 68.48 |
| 3% | MGW | 98.80 | 96.77 | 95.15 | 65.56 | 95.38 | 93.54 | 89.94 | 68.78 |
| 6% | Masking | 98.71 | 96.25 | 91.04 | 63.78 | 95.26 | 92.32 | 88.26 | 67.53 |
| 6% | Interpolation | 98.89 | 97.98 | 95.09 | 68.41 | 97.10 | 95.80 | 92.85 | 71.48 |
| 6% | MGW | 98.90 | 97.94 | 95.20 | 70.75 | 97.09 | 95.78 | 93.09 | 74.09 |
| 9% | Masking | 99.10 | 98.01 | 93.18 | 69.22 | 95.91 | 93.96 | 90.17 | 69.81 |
| 9% | Interpolation | 99.22 | 98.35 | 95.42 | 72.35 | 96.30 | 94.51 | 92.48 | 72.57 |
| 9% | MGW | 99.18 | 98.41 | 95.58 | 74.01 | 96.32 | 94.53 | 92.67 | 74.49 |

variation in the *MGW-Loss* function. We also show that the Gaussian weighted interpolation gives best performance in table 1 and 2.

## C.1 Selection of sigma ($\sigma$)

To further illustrate the effects of varying values of sigma ($\sigma$) in the proposed *MGW-Loss* function, we evaluate the same loss function at different sigmas in table 3. We show different weight distributions for various sigma values in figure 2, with the loss behaving as masking loss at very low sigma ($\sigma < 0.01$) and as interpolation loss at high sigma ($\sigma > 5$). Since *MGW-Loss* will consider weights for the interpolation based pseudo-labeled neighboring frames based on their distance from the annotated frame, the value of $\sigma$ will affect it significantly. Low sigma will give little to no weight to neighbor frames while very high sigma will give all pseudo-labeled frames equal weights, which might be incorrect with sparse annotations. Based on this, we prefer to pick a moderate sigma value that gives partial weight to neighboring frames. We conduct our experiments with sigma values of $\sigma = 1.3$ and $\sigma = 2.5$ in table 3. It is found that a leaner distribution using $\sigma = 1.3$ gives better initialization with sparse frames than $\sigma = 2.5$ since interpolated annotations are not as trustworthy in the beginning. Due to a stronger initialization, $\sigma = 1.3$ stays ahead and gives best scores for higher annotation percentages as well.

## C.2 Influence of Lambda $\lambda$

We perform an experiment to study the influence of $\lambda$ on Eq. 3. The effect of different values of $\lambda$ is shown in table 4. We observe that giving higher weight to uncertainty ($\lambda = 0.75$) reduces overall score as it ignores proximity value. Having lower weight on uncertainty ($\lambda = 0.25$) on the other hand promotes more distance and performs better than $\lambda = 0.75$. Although $\lambda = 0.25$ has higher f-mAP @ 0.5, we do not optimize this hyperparameter and use equal weight in the paper ($\lambda = 0.5$).

Table 3: Results of using various values for $\sigma$ in the proposed *MGW-Loss* in UCF-101. The effect of *MGW-Loss* changes for different values of $\sigma$, with it behaving as masking loss at $\sigma = 0$, interpolation loss at $\sigma >= 5$ and *MGW* loss at $0 < \sigma < 5$.

| Annotation Percent | Sigma $\sigma$ | v-mAP@ | | | | f-mAP@ | | | |
|---|---|---|---|---|---|---|---|---|---|
| | | 0.1 | 0.2 | 0.3 | 0.5 | 0.1 | 0.2 | 0.3 | 0.5 |
| 1% | 0 | 97.89 | 93.45 | 86.53 | 59.4 | 91.04 | 88.25 | 83.54 | 60.1 |
| 1% | 1.3 | 98.07 | 95.37 | 89.01 | 63.94 | 91.28 | 88.41 | 83.85 | 64.69 |
| 1% | 2.5 | 98.01 | 95.17 | 89.00 | 63.79 | 91.20 | 88.35 | 83.69 | 64.38 |
| 1% | 5 | 97.99 | 94.24 | 87.69 | 63.71 | 91.14 | 88.45 | 83.69 | 64.04 |
| 5% | 0 | 98.01 | 95.1 | 89.98 | 67.6 | 93.18 | 90.07 | 87.78 | 66.50 |
| 5% | 1.3 | 98.12 | 96.01 | 90.95 | 71.89 | 93.38 | 91.28 | 88.71 | 70.91 |
| 5% | 2.5 | 98.1 | 95.89 | 90.78 | 71.55 | 93.3 | 91.25 | 88.56 | 70.26 |
| 5% | 5 | 98.05 | 95.3 | 90.28 | 69.41 | 93.21 | 91.3 | 88.16 | 68.41 |
| 10% | 0 | 98.10 | 95.84 | 90.23 | 69.9 | 94.02 | 92.05 | 88.30 | 69.30 |
| 10% | 1.3 | 98.14 | 96.46 | 91.12 | 73.2 | 94.05 | 92.03 | 88.72 | 71.75 |
| 10% | 2.5 | 98.10 | 96.45 | 91.11 | 73.07 | 94.05 | 92.02 | 88.68 | 71.59 |
| 10% | 5 | 98.18 | 96.43 | 91.12 | 71.52 | 94.07 | 92.07 | 88.10 | 70.57 |

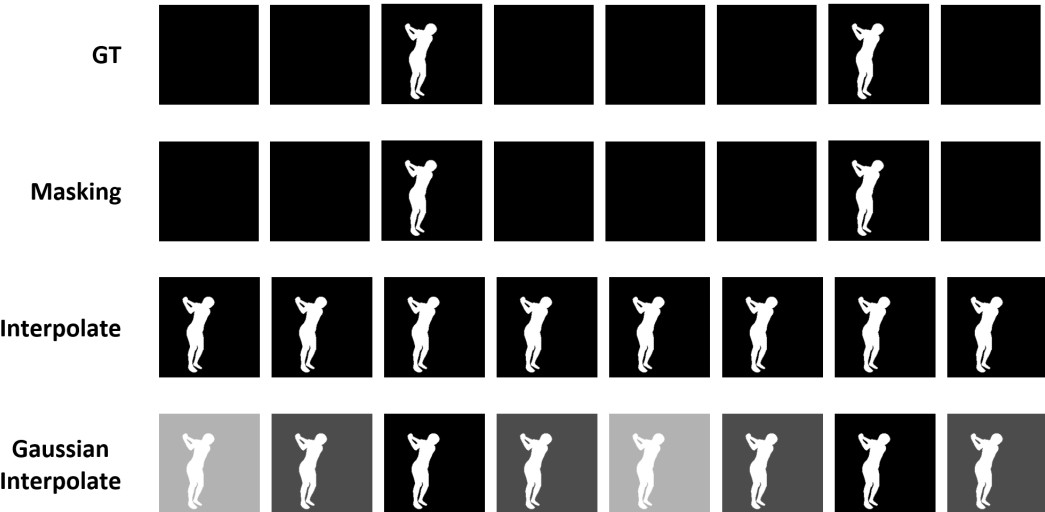

Figure 1: Representation of using different $\sigma$ values for *MGW-Loss* computation. **Row 1:** Ground truth annotation (sparse) for a single video clip. **Row 2:** *MGW* with $\sigma = 0$ (equivalent to masking out frames with GT present for loss computation). **Row 3:** *MGW* with high $\sigma$ (High weight given to all neighboring frames, equivalent to using interpolated ground truth annotations). **Row 4:** *MGW* with $\sigma = 1.3$ (Max Gaussian value used as weight for each interpolated annotation).

## C.3  Influence of Tau $\tau$ in non-activity suppression

As the model is very certain on most background pixels early on in the training, we observed that this gives us a large margin between the true background pixel confidence value and the uncertain pixel confidence value. We perform ablation with different $\tau$ values and show them in table 4. We observe that changing $\tau$ affects the results slightly and hyperparameter tuning can affect the final results with small margin. We did not perform hyperparameter tuning for $\tau$ in our proposed method.

## D  Comparison to prior supervised approach

We compare our proposed approach with prior supervised methods in Table 5 on UCF-101 and J-HMDB dataset. Using only 10% of annotation on both datasets, our *APU+MGW-Loss* based approach performs comparably with the same backbone using 100% annotation. Our 10% model's

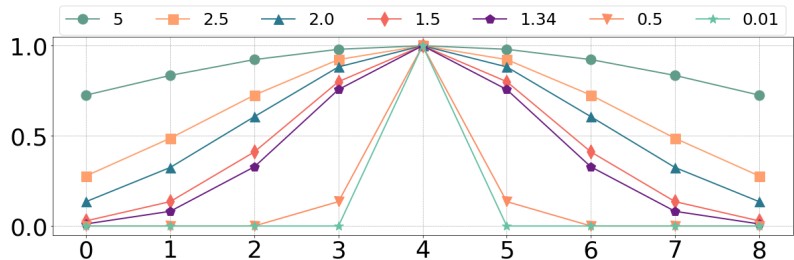

Figure 2: Weight distribution for different values of sigma $\sigma$ for the *MGW-Loss* function. Horizontal axis shows the frames for a sample clip (with ground truth annotation at frame 4) and vertical axis shows the weights given to each frame based on *MGW-Loss* with varying $\sigma$. The weights for $\sigma$ from 0.01 till 5 is shown here, with $\sigma < 0.01$ acting as masking loss and $\sigma > 5$ acting as interpolation loss and the values in between representing *MGW* loss.

Table 4: Influence of $\lambda$ and $\tau$ in the training. We evaluate the effect of using different value of $\lambda$ and $\tau$ on UCF-101 using the proposed *APU* to increase annotations from 5% to 10%.

| Lambda | v-mAP@ | | f-mAP@ | | Tau | v-mAP@ | | f-mAP@ | |
|---|---|---|---|---|---|---|---|---|---|
| ($\lambda$) | 0.3 | 0.5 | 0.3 | 0.5 | ($\tau$) | 0.3 | 0.5 | 0.3 | 0.5 |
| 0.5 | 91.12 | 73.20 | 88.72 | 71.75 | 0.3 | 90.96 | 72.42 | 88.24 | 71.65 |
| 0.25 | 92.34 | 72.59 | 88.65 | 71.83 | 0.4 | 91.12 | 73.20 | 88.72 | 71.75 |
| 0.75 | 90.99 | 72.09 | 88.06 | 71.67 | 0.5 | 91.24 | 73.15 | 88.55 | 71.77 |

f-mAP@0.5 and v-mAP@0.5 is closer to multiple prior supervised models and can be trained with a simpler end-to-end method.

Table 5: Comparison with prior state-of-the-art supervised methods on UCF-101 and J-HMDB. We evaluate our full approach on v-mAP and f-mAP scores using only 10% annotation data. **f@** denotes **f-mAP@**. † is trained in weakly-supervised manner.

| Method | Annot Percent | UCF-101 | | | | | | J-HMDB | | | | |
| | | f@ | v-mAP@ | | | | | f@ | v-mAP@ | | | |
| | | 0.5 | 0.1 | 0.2 | 0.3 | 0.5 | | 0.5 | 0.1 | 0.2 | 0.3 | 0.5 |
|---|---|---|---|---|---|---|---|---|---|---|---|---|
| Peng et al. [1] | 100% | 65.7 | 77.3 | 72.9 | 65.7 | 35.9 | | 58.5 | - | 74.3 | - | 73.1 |
| TCNN [2] | 100% | 67.3 | 77.9 | 73.1 | 69.4 | - | | 61.3 | - | 78.4 | - | - |
| Gu et al. [3] | 100% | 76.3 | - | - | - | 59.9 | | 73.3 | - | - | - | - |
| ACT [4] | 100% | 69.5 | - | 76.5 | - | - | | - | - | 74.2 | - | 73.7 |
| STEP [5] | 100% | 75.0 | 83.1 | 76.6 | - | - | | - | - | - | - | - |
| Rel. Graph [6] | 100% | 77.9 | - | - | - | - | | - | - | - | - | - |
| AIA [7] | 100% | 78.8 | - | - | - | - | | - | - | - | - | - |
| VidsCapsNet [8] | 100% | 78.6 | 98.6 | 97.1 | 93.7 | 80.3 | | 64.6 | 98.4 | 95.1 | 89.1 | 61.9 |
| Ours | 100% | 74.0 | 98.3 | 96.9 | 91.5 | 75.2 | | 74.9 | 99.2 | 99.2 | 96.4 | 75.8 |
| Ours † | 10% | 71.7 | 98.1 | 96.5 | 91.1 | 73.2 | | 74.5 | 99.2 | 98.4 | 95.6 | 74.0 |

# E   Frame selection details

## E.1   Uncertainty using dropout

We follow [9] to use dropout for generating uncertainty during inference. We apply *3D Dropout* with a dropout probability of $0.5$ before the final convolution layer in the decoder network from [8]. This dropout layer is enabled during inference, enabling uncertainty in the output by avoiding memorization. We pass the same clip $10$ times in the network with dropout enabled and collect the output for uncertainty based scoring for each frame.

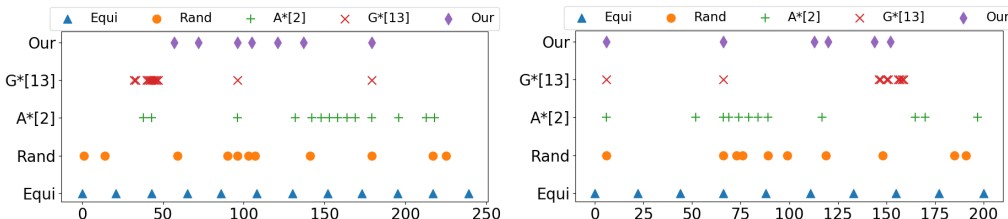

Figure 3: Visualization of frames selected using different cost functions for **(Left)** Soccer Juggling, **(Right)** Dog Walking. We show results for Our, random, equidistant, G*: Gal et al. [9] and A*: Aghdam et al. [10], with all methods using same detection backbone for fair comparison. We visualize two videos with long continuous actions, where our global ranking based frame selection approach selects only few frames for annotation which are well spaced. Both baseline active learning methods (G*, A*) end up selecting more frames since they don't have global ranking in their formulation. Non-active learning baseline (random, equidistant) select frames without any utility information. Our method has better cost utilization by reducing redundant samples while maintaining better performance than any other baseline method.

Table 6: Comparison between frame selection with different step size in UCF-101. For each setting, we use the *MGW-Loss* with $\sigma = 1.3$ for consistency. We use the the proposed **APU** frame selection method. We start with an initial random seed annotation and increase annotations by given step size in each iteration.

| Annotation | v-mAP@ | | | | f-mAP@ | | | |
|---|---|---|---|---|---|---|---|---|
| Percent | 0.1 | 0.2 | 0.3 | 0.5 | 0.1 | 0.2 | 0.3 | 0.5 |
| **Step size 1%** | | | | | | | | |
| 1% | 98.07 | 95.37 | 89.01 | 63.94 | 91.28 | 88.41 | 83.85 | 64.69 |
| 2% | 98.07 | 95.41 | 89.42 | 66.4 | 92.14 | 88.65 | 84.20 | 67.10 |
| 3% | 98.08 | 95.68 | 90.01 | 69.52 | 93.84 | 89.05 | 86.11 | 69.02 |
| 4% | 98.12 | 95.89 | 90.72 | 71.00 | 94.04 | 89.89 | 87.51 | 70.00 |
| 5% | 98.12 | 96.10 | 91.03 | 72.20 | 94.10 | 91.35 | 88.62 | 70.99 |
| 6% | 98.15 | 96.18 | 91.03 | 72.70 | 94.12 | 91.52 | 88.73 | 71.31 |
| 7% | 98.14 | 96.27 | 91.12 | 73.05 | 94.10 | 91.68 | 88.85 | 71.55 |
| 8% | 98.15 | 96.32 | 91.15 | 73.21 | 94.14 | 91.84 | 88.80 | 72.47 |
| 9% | 98.19 | 96.45 | 91.22 | 73.82 | 94.15 | 92.10 | 88.84 | 72.76 |
| 10% | 98.22 | 96.58 | 91.22 | 73.94 | 94.15 | 92.10 | 88.89 | 72.91 |
| **Step size 5%** | | | | | | | | |
| 1% | 98.07 | 95.37 | 89.01 | 63.94 | 91.28 | 88.41 | 83.85 | 64.69 |
| 5% | 98.12 | 96.01 | 90.95 | 71.89 | 93.38 | 91.28 | 88.71 | 70.91 |
| 10% | 98.14 | 96.46 | 91.12 | 73.20 | 94.05 | 92.03 | 88.72 | 71.75 |

## E.2   Selection step size

Another crucial factor that can affect learning process initially is the frame selection step size. This is directly related to the time needed for training. Larger step size will take shorter time as fewer selection iterations are needed to expand the dataset while shorter step size will need multiple selection iterations and can take longer as a result. Table 6 shows the difference of slow increment vs fast increment vs random selection. While doing a larger step size of 5%, our final result is already significantly better than random selection. When we perform annotation improvement for smaller step size of 1%, we notice that the performance is consistently better than larger step size. With a slow step size, our 10% data variant significantly outperforms both 10% with large step size and 10% random frame selection.

## E.3   Local vs Global frame selection

Frame selection can be done locally per each video or globally across all videos. While local frame selection will select top K% frames with highest utility score for a given video, it might

Table 7: Comparison between different frame selection method (Global selection vs Local selection) in JHMDB. For each setting, we train the model using the *MGW-Loss* with $\sigma = 1.3$.

| Annotation | v-mAP@ | | | | f-mAP@ | | | |
|---|---|---|---|---|---|---|---|---|
| Percent | 0.1 | 0.2 | 0.3 | 0.5 | 0.1 | 0.2 | 0.3 | 0.5 |
| **Global** | | | | | | | | |
| 3% | 98.81 | 96.77 | 95.15 | 65.56 | 95.38 | 93.54 | 89.94 | 68.78 |
| 6% | 98.97 | 97.94 | 95.2 | 70.75 | 97.09 | 95.78 | 93.09 | 74.09 |
| 9% | 99.18 | 98.41 | 95.58 | 74.01 | 96.32 | 94.53 | 92.67 | 74.50 |
| **Local** | | | | | | | | |
| 3% | 98.81 | 96.77 | 95.15 | 65.56 | 95.38 | 93.54 | 89.94 | 68.78 |
| 6% | 98.82 | 97.12 | 95.18 | 69.44 | 96.21 | 94.04 | 91.87 | 72.81 |
| 9% | 99.15 | 98.08 | 95.19 | 72.83 | 96.25 | 94.73 | 92.48 | 73.55 |

Table 8: Per class score for Global vs Local frame selection method on JHMDB dataset. We report the number of frames selected and the score for video level AP and frame level AP at IoU @ 0.5 for global selection method and local selection method.

| Class | Frame selection | | Global scores | | Local scores | |
|---|---|---|---|---|---|---|
| | Global | Local | video-AP | frame-AP | video-AP | frame-AP |
| brush_hair | 391 | 232 | 0.75 | 0.68 | 0.67 | 0.60 |
| catch | 155 | 272 | 0.21 | 0.14 | 0.00 | 0.06 |
| clap | 382 | 248 | 1.00 | 0.90 | 0.85 | 0.86 |
| climb_stairs | 123 | 224 | 0.00 | 0.00 | 0.00 | 0.06 |
| golf | 240 | 240 | 0.00 | 0.10 | 0.00 | 0.05 |
| jump | 68 | 216 | 0.08 | 0.14 | 0.08 | 0.12 |
| kick_ball | 72 | 200 | 0.00 | 0.00 | 0.00 | 0.00 |
| pick | 110 | 224 | 0.08 | 0.22 | 0.33 | 0.24 |
| pour | 404 | 312 | 0.94 | 0.89 | 1.00 | 0.87 |
| pullup | 405 | 312 | 0.19 | 0.20 | 0.13 | 0.22 |
| push | 108 | 240 | 0.17 | 0.22 | 0.08 | 0.15 |
| run | 89 | 232 | 0.27 | 0.27 | 0.27 | 0.25 |
| shoot_ball | 133 | 232 | 0.00 | 0.02 | 0.00 | 0.01 |
| shoot_bow | 458 | 304 | 1.00 | 0.72 | 1.00 | 0.89 |
| shoot_gun | 343 | 312 | 0.25 | 0.30 | 0.25 | 0.19 |
| sit | 284 | 216 | 0.58 | 0.53 | 0.58 | 0.52 |
| stand | 254 | 200 | 0.64 | 0.65 | 0.64 | 0.68 |
| swing_baseball | 332 | 312 | 0.00 | 0.05 | 0.00 | 0.17 |
| throw | 430 | 280 | 0.27 | 0.17 | 0.18 | 0.15 |
| walk | 166 | 232 | 0.17 | 0.14 | 0.17 | 0.18 |
| wave | 333 | 240 | 0.92 | 0.93 | 0.67 | 0.66 |

not take into account similar frames across other videos for same class. This poses a possibility for redundant information in the annotation. Since our goal is to reduce annotation budget while increasing information for fewest annotations possible, we want to utilize our annotation budget wisely. A possible solution is to globally rank the frames based on utility across all videos, while still maintaining distance based scoring. We propose the global frame selection process, where we score the frames for each video based on Gaussian distance from closest annotated frame in that video, but rank them globally across all videos. Once a frame is selected, we update the score for all other frames nearby based on distance and re-rank them. The score updating is only done for distance based factors of the equation and does not need running of the frames through the model again, reducing any significant extra time compared to local frame selection. Table 7 shows the difference between doing local selection vs global selection, where the network learns better from annotations curated using the global selection method. We further evaluate the per class score for frame selection method in table 8. We can see the difference in number of frames selected for each class in global and local method, which shows that global selection prioritizes frame selection over certain classes while local selection tries to pick same number of sample for each of the video. We

Table 9: Comparison between the proposed frame selection method and sample selection method in UCF-101. Sample selection method selects entire video for annotation at the given percentage.

| Annotation | v-mAP@ | | | | f-mAP@ | | | |
| Percent | 0.1 | 0.2 | 0.3 | 0.5 | 0.1 | 0.2 | 0.3 | 0.5 |
| **Our (Frame)** | | | | | | | | |
| 1% | 98.07 | 95.37 | 89.01 | 63.94 | 91.28 | 88.41 | 83.85 | 64.69 |
| 5% | 98.12 | 96.01 | 90.95 | 71.89 | 93.38 | 91.28 | 88.71 | 70.91 |
| 10% | 98.14 | 96.46 | 91.12 | 73.20 | 94.05 | 92.03 | 88.72 | 71.75 |
| **Sample** | | | | | | | | |
| 1% | 89.48 | 76.81 | 61.02 | 23.42 | 76.47 | 69.37 | 58.64 | 31.65 |
| 5% | 91.81 | 82.88 | 70.85 | 36.24 | 81.41 | 75.02 | 66.43 | 42.68 |
| 10% | 95.94 | 88.89 | 79.35 | 48.21 | 86.71 | 81.33 | 73.87 | 50.37 |

Table 10: Comparison between the proposed frame selection method and sample selection method in JHMDB. Sample selection method selects entire video for annotation at the given percentage.

| Annotation | v-mAP@ | | | | f-mAP@ | | | |
| Percent | 0.1 | 0.2 | 0.3 | 0.5 | 0.1 | 0.2 | 0.3 | 0.5 |
| **Our (Frame)** | | | | | | | | |
| 3% | 98.81 | 96.77 | 95.15 | 65.56 | 95.38 | 93.54 | 89.94 | 68.78 |
| 6% | 98.97 | 97.94 | 95.2 | 70.75 | 97.09 | 95.78 | 93.09 | 74.09 |
| 9% | 99.18 | 98.41 | 95.58 | 74.01 | 96.32 | 94.53 | 92.67 | 74.50 |
| **Sample** | | | | | | | | |
| 3% | 87.36 | 86.94 | 81.05 | 47.42 | 87.98 | 86.22 | 80.11 | 50.09 |
| 6% | 92.05 | 90.45 | 88.24 | 53.69 | 90.36 | 89.74 | 85.88 | 56.32 |
| 9% | 94.15 | 92.18 | 90.76 | 57.33 | 93.68 | 91.85 | 90.30 | 58.85 |

observe that for most classes which global selection picks more frames from are scoring higher than the local selection counterpart.

### E.4 Frame vs Sample selection

We evaluate the alternate sampling technique in which we start with only a fraction of videos fully annotated and select the videos to annotate in each active learning step. This is different to our frame selection strategy where we assume all videos have a fraction of frames annotated and we only select which frames to annotate from all videos in each active learning step. Sample selection is a more costly approach as each selected video has to be fully annotated for all frames, which adds annotation cost for little gain in knowledge. We show the results for UCF-101 and JHMDB dataset in table 9 and 10 respectively for sample selection vs our frame selection method.

### E.5 Extending annotation to convergence

We extend the baselines as well as our method further in Figure 4, where we see that the baselines eventually converge close to fully-supervised scores with additional annotations. Our *APU* frame selection approach gets higher score earlier and saturates from 20% annotation while the random baseline catches up at around 40% annotation, demonstrating *APU*'s need for less annotation to achieve higher scores.

### E.6 Cost function variations

Frame selection can be done through different scoring mechanisms such as random selection, uncertainty based selection or our *Adaptive Proximity-aware Uncertainty (APU)* based selection. Each selection method gives different set of frames for further annotation to the oracle. Our goal is to select frames with maximum utility for further annotation such that the network can learn better from it. Random frame selection works as the baseline method, where frames are selected at random for annotation. We compare that against uncertainty based selection method, where each frame is scored based on the uncertainty score as [9]. While this method highlights regions that are more uncertain

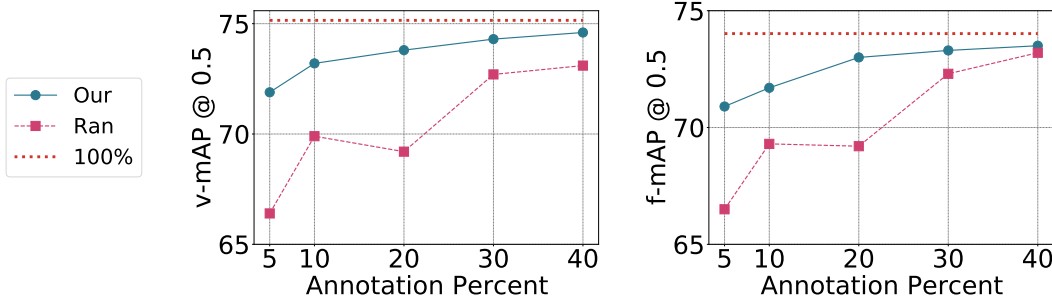

Figure 4: Extending random baseline and our *APU* approach till convergence for UCF-101 (v-mAP, f-mAP @ 0.5 IoU).

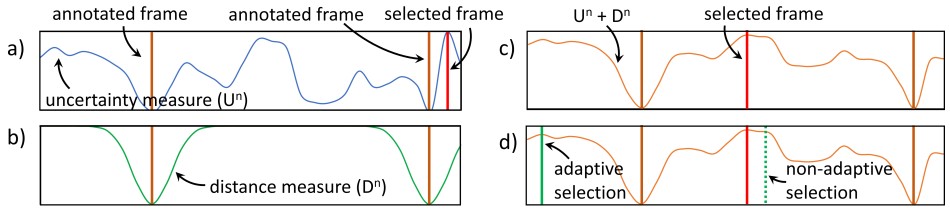

Figure 5: Active frame selection using APU scoring function. a) frame selection based on just uncertainty, b) showing distance measure based on existing annotations, c) frame selection using proposed APU scoring function, and d) difference between frame selection with adaptive and non-adaptive scoring.

---

**Algorithm 1** Active Sparse Learning (ASL) algorithm

---

    **Input:** videos $\mathcal{V}$; total budget $b$; labelled frames $F_L$; number of trails $T$; pixels per frame $I^p$; current model $\mathcal{M}^c$; frame annotation cost $C_{frame}$

    **Initialize:** budget $b_s = 0$; frames to annotate $F_{annot} \leftarrow \{\}$

1: **for all** videos $v$ in $\mathcal{V}$ **do**
2:     **for all** frame $i$ in $f_{v,u}$ **do**     *// Iterate over each unannotated frame*
3:         $Uncertainty(U^i) = U(\mathcal{M}^c, i)$ *// Compute uncertainty of frame $i$ with Eq. 1*
4: **while** $b_s \leq b$ **do**
5:     APU_score $\leftarrow \{\}$     *// Initialize empty list to store all frame's APU score*
6:     **for all** videos $v$ in $\mathcal{V}$ **do**
7:         **for all** frame $i$ in $f_{v,u}$ **do** *// Iterate over each unannotated frame*
8:             $D^i = D(f_{v,l}, i)$     *// Compute distance score of frame i using Eq. 2*
9:             $\mathcal{U}^i_{APU} = (U^i, D^i)$     *// Compute frame i utility score using Eq. 3*
10:            Append $\mathcal{U}^i_{APU}$ to APU_score
11:     $f_{max} \leftarrow \max(APU\_score)$     *// Get frame with highest utility*
12:     Append $f_{max}$ to $F_{annot}$     *// Add highest utility frame for annotation*
13:     $F_L \leftarrow F_L \cup f_{max}$ *// Mark frame as annotated for future distance scores*
14:     $b_s \leftarrow b_s + C_{frame}$ *// Increment cost for each frame added to annotation*
15: **return** $F_L, F_{annot}$     *// Return new labelled frames and set for annotation*

---

and need extra annotation, it fails to take into account the temporal factor for the scoring process and ends up selecting frames from similar region. Our *APU* based method adapts to each new annotation and scores nearby frames lower, thus encouraging frame selection from different regions. Figure 5 demonstrates how the proposed *APU* selection method picks optimum frames compared to other techniques. We compare sample selection using *APU* and other baseline methods in figure 3. Unlike uncertainty based method, our *APU* based selection method selects frames further from each other and allows more variation in annotation.

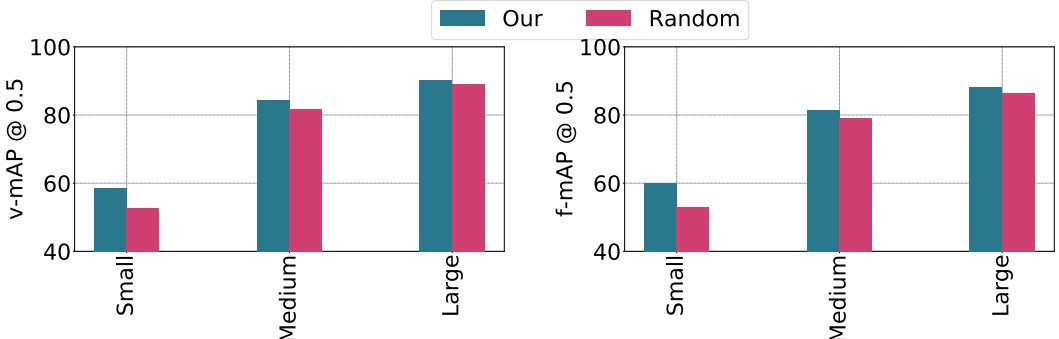

Figure 6: Evaluation of performance of the model with respect to relative area of the spatio-temporal annotation.

### E.7 Selection algorithm

We describe our selection algorithm from the main manuscipt in algorithm 1. Once we have a model trained ($\mathcal{M}^c$), we iterate over each video and get the uncertainty score for each unannotated frame. This step requires the model output so we run it once for all the videos for each active learning cycle. Then we need to compute the distance score for each frame, combine it with the uncertainty score for that frame and then pick the frame among all videos with the highest utility. Since the distance score computation has to be recomputed for each video which had a new annotation frame added, we also recompute the total utility score for those videos. However, as the uncertainty score does not need to be recomputed, this process is fairly fast. We keep on adding frame for annotation until we exhaust the budget for that active learning round. At the end we return a set of frames selected based on their utility for the oracle to annotate and add to labeled set.

### E.8 Performance based on annotation area

Similar to MS-COCO dataset, we also separated the evaluation set into small, medium and large based on each video's average activity area with respect to the frame size [Small < 702, Medium >= 702 and < 1302, Large >= 1302 in square pixels for UCF-101]. With this distinction, we had 267 small videos, 479 medium videos and 164 large videos. As all video classes don't have all three variations, some of the classes don't exist in small and large sets. As such, we only compare the common classes across all three sets for equivalent comparison. From Figure 6, we see that our approach selects samples to improve small and medium sized actions more compared to random selection baseline

## F   VOS task

We demonstrate the effectiveness of our approach on video object segmentation task using [11] and [12] networks. We show the results for [12] network in table 11, where we compare with baseline active learning methods using random, uncertainty-based [9], entropy-based [10] with our proposed active learning method. We observe that the overall score in each incremental step is higher for the frames selected by our active learning method compared to all baselines, further validating the generalization of our approach on different video understanding task.

## G   Network implementation details

### G.1   Action detection network

We use the 2D variant of video capsule network [8] for action detection task on UCF-101-24 and JHMDB-21 dataset. The network takes an input clip of $T \times H \times W \times C$ dimension [$T$=frames, $H$=height, $W$=width, $C$=channels] and outputs $T$ frames of $H \times W \times 1$ dimension. It also predicts the class prediction vector for the entire clip. The 2D capsule network takes a batch size of 8 samples

Table 11: Comparison of the proposed method on YouTube-VOS 2019 dataset with baseline active learning methods using LSTM based VOS network from [12]. R = Random, A = Aghdam et al. [10], G = Gal et al. [9]. * is extended to video object segmentation using same segmentation network as [12].

| Method | Overall | | | $\mathcal{J}_S$ | | | $\mathcal{J}_U$ | | | $\mathcal{F}_S$ | | | $\mathcal{F}_U$ | | |
|---|---|---|---|---|---|---|---|---|---|---|---|---|---|---|---|
| | 10% | 20% | 30% | 10% | 20% | 30% | 10% | 20% | 30% | 10% | 20% | 30% | 10% | 20% | 30% |
| Random | 20.9 | 24.5 | 26.3 | 26.6 | 30.2 | 33.4 | 20.1 | 23.4 | 24.5 | 19.8 | 24.1 | 25.9 | 17.0 | 20.5 | 21.4 |
| A *[10] | 22.7 | 26.7 | 34.8 | 30.4 | 34.2 | 41.8 | 20.3 | 23.2 | 30.0 | 22.8 | 28.4 | 39.7 | 17.6 | 21.1 | 27.6 |
| G * [9] | 21.7 | 26.0 | 33.3 | 30.2 | 30.4 | 41.2 | 19.7 | 20.3 | 30.2 | 21.0 | 22.8 | 34.4 | 15.9 | 17.6 | 27.3 |
| Our | 24.3 | 33.3 | 38.1 | 31.5 | 41.2 | 45.7 | 22.5 | 30.2 | 32.1 | 24.0 | 34.4 | 42.9 | 19.3 | 27.3 | 31.6 |

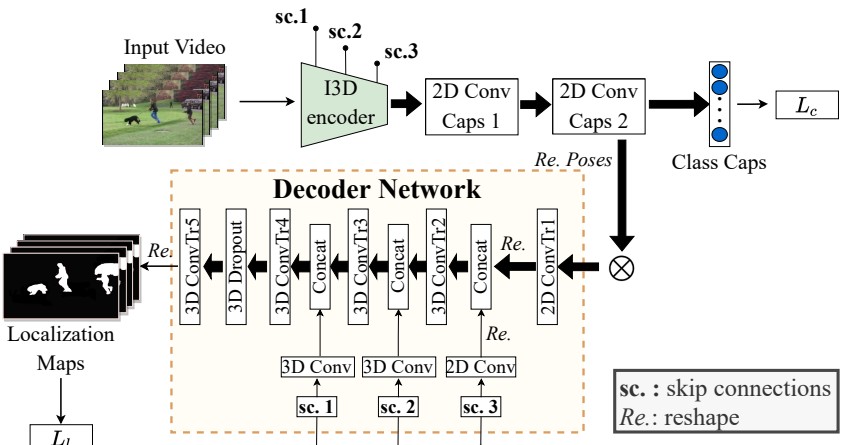

Figure 7: Overview of the proposed action detection network. Based on [8], features are extracted from input frames using I3D [13] architecture based encoder. We take features from $Mixed\_4f$ layer of I3D network. This is then followed by two $2D\ convolutional\ capsule$ layers which outputs class capsules. The class capsules is used for final class prediction and classification loss computation. This is followed by series of transpose convolution layers (2D and 3D) for upscaling the feature map and concatenation with features from intermediate layers of the I3D encoder via skip connections. We finally obtain the localization maps of same size as input video, which is used for detection loss.

per iteration, with each sample clip of size $8 \times 224 \times 224 \times 3$ with a temporal skip rate of 2. We follow the same input/output format as the original paper for 3D capsule network [8] for the 2D network variant. The full architecture detail is shown in figure 7.

## G.2   VOS network

We use our technique on two separate newtorks for the video object segmentation task. We use the method by [11] that recently had state-of-the-art performance on YouTube-VOS 2019 challenge. This network takes an input clip of 3 frames and learns up-to 2 objects segmentation across those 3 frames in training time. We also test our method in a simpler network used for VOS task based on ConvLSTM modules [12]. It takes as input a clip of $T$ frames of $H \times W \times C$ dimension and the annotation mask of first frames of the clip of size $H \times W \times 1$. The network outputs predicted segmentation mask of size $T \times H \times W \times C_{class}$. For both networks we use clips with frames of size $224 \times 224 \times 3$ and annotation of size $224 \times 224 \times 1$ as input and the networks output segmentation clip with size $224 \times 224 \times 1$. The first method [11] takes 3 frames input in the clip while the second method [12] takes 32 frames input in the clip. The object of interest annotated in the first frame will be segmented through the clip. During inference, we use the annotation of the last frame predicted as the input annotation mask for the next subsequent clip. We use a batch size of 8 for this task as well.

Table 12: Comparison between Kinetics pre-trained and Charades pre-trained weights for UCF-101 dataset. We show the f-mAP and v-mAP scores at 0.5 IoU for different annotation size.

| Pre-train | f-mAP @ 0.5 | | | v-mAP @ 0.5 | | |
|---|---|---|---|---|---|---|
| | 1% | 5% | 10% | 1% | 5% | 10% |
| Charades | 60.7 | 66.5 | 69.3 | 59.2 | 66.4 | 69.9 |
| Kinetics | 60.4 | 66.5 | 69.0 | 59.2 | 66.2 | 69.8 |

### G.3 Technical details

We train our model using a single *16GB Nvidia RTX 5000 GPU* with Turing architecture. The frame selection method only runs in inference mode with *Dropout* enabled, thus using only a fraction of the GPU memory. Due to this, we can run multiple instances in parallel for frame selection in the training video set, reducing the time taken for frame selection process. During each iteration, we only select the given percentage of frames for further annotation and we retain the previous set of annotated frames. On a 8 core 3.2 GhZ Intel CPU and 16GB Nvidia RTX 5000 GPU combination the frame selection round takes **50 minutes** for UCF-101 (432K training frames). We run a total of 2 active learning cycles for frame selection in the main experiments (1% -> 5% and 5% -> 10%). The model training for UCF-101 takes about **15 minutes** per epoch, which is trained for 40 epochs for each set of annotations.

### G.4 Interpolation:

The annotation interpolation for UCF-101 is done using linear interpolation of the bounding box corners. The pixel-wise annotation interpolation for J-HMDB is done using CyclicGen [14]. In case of edge frames or single frame annotations, we extrapolate the annotation to other frames.

### G.5 Pre-trained weights:

While we follow prior works and use Charades pre-trained weights in our experiments, we also compare with Kinetics-400 pre-trained weights in table 12. This shows that the pre-trained weights selection will not affect the trend from *APU* selection.