# OpenReview forum: "Are all Frames Equal? Active Sparse Labeling for Video Action Detection"
_NeurIPS.cc/2022/Conference — NeurIPS 2022 Accept_

### Official Review · Reviewer_mzzJ · 2022-07-08

**Rating:** 7
**Confidence:** 5
**Soundness:** 3 good
**Presentation:** 4 excellent
**Contribution:** 3 good

**Summary:**

This paper describes a novel training paradigm for action detection in videos.

Technically, the approach consists in an sparse labelling model that implements a frame-level scoring module that tries to select the most informative/discriminative frames for action detection. With this sparse labels, the paper describes a training approach, with its associated loss function.

An experimental evaluation is performed in two publicly available datasets, and the results reveal the benefits of the sparse active labeling technique.

**Questions:**

This is a paper that may be of interest to the computer vision community. The manuscript is well written, and the ideas they propose are novel (there is no similar work using AL for action detection in videos). The experimental results are convincing and allow us to judge the contributions claimed from the paper. In addition, the experimental evaluation developed is thorough and critical. Overall, except for some minor shortcomings that I would like the authors to address, I believe the article should be accepted.

**Limitations:**

Yes. The authors identified as the main limitation of the approach that as most AL is a time consuming model. Clarifications about the runtime are needed, as I pointed above.

**Strengths And Weaknesses:**

# Strengths

- Writing a scientific article is not easy, but writing it well is really an art. This manuscript has been written in a careful and engaging way for the reader. Ideas are not masked in confusing paragraphs, but are clearly explained. It's a pleasure to be touched to review articles like this. I thank the authors for their efforts.

- Section 2 adequately discusses previous works, showing the novelty of the proposed approach. Overall, the application of active sparse labeling for training action detection models in videos has not been previously explored.

- The experimental evaluation follows clear experimental setups using publicly available datasets (UCF-101-24 and J-HMDB-21). The comparison with state-of-the-art models is fair and sound.

- Active Sparse Labeling idea is worth to be shared with the rest of the computer vision community. The way this paper treats the sparse labels could be of interest for continual learning approaches, where models face to new tasks for which no annotations (or a few) could be at hand. Moreover, the learning with the Max-gaussian weighted loss could be applied to other semi-supervised pipelines, where the loss can offer a mechanism for assigning to the pseudo-labels a sort of confidence measure.

- The experimental evaluation is well designed: specific experiments to validate the contributions claimed in the paper. It also reports comparisons with state-of-the-art semi-supervised methods, and a thorough ablation study. The paper defines the state-of-the-art for the semi-supervised setting on the problem, with just 10% of the frames being annotated with ground-truth.



# Weaknesses


- As it is pointed in the manuscript, as an AL model, scalability is a weakness of the model. We need multiple iterations to select the frames, and this is time consuming. It would be fundamental to know how much time needs the deep learning architecture between iterations.

- I found some technical limitations that are worth to be discussed:
a) It was not clear to me why does the model need an estimation of the uncertainty per pixel, that has to be obtained utilizing MC-Dropout. To run Eq. 1 should be highly time-consuming. Are there any other uncertainty measures that could be used? Have the authors used any alternative?
b) Adaptive Proximity-aware Uncertainty (APU) objective is to select frames with temporal diversity, am I wrong? Eq. 3 presents a combination between the distance and the uncertainty with just a sum. Are both variables scaled? With a lambda of 0.5 the model assumes that the two terms are in the same range of values. An ablation study on the influence of lambda could be interesting.
c) In the Non-activity suppression block, I wonder the influence of tau in the performance of the model.
d) It is unclear to me how the proposed model is integrated in VideCapsuleNet [9] approach. Section 3.e needs to be extended.

# Minor comments:
- Please, punctuate all the equations. They are part of the text.

---

> ### Author Response · Authors · 2022-08-02
> **Response to reviewer mzzJ (part 1 of 2)**
>
> We sincerely thank the reviewer for the valuable feedback and analysis on the paper. We have addressed the questions and concerns raised by the reviewer.
>
> **W.1 As it is pointed in the manuscript, as an AL model, scalability is a weakness of the model. We need multiple iterations to select the frames, and this is time consuming. It would be fundamental to know how much time the deep learning architecture needs between iterations.**
>
> We agree with the reviewer that scalability is in general a weakness of active learning. For our hardware setting (8 core 3.2 GhZ Intel CPU and a 16GB Nvidia RTX 5000 GPU) each training cycle took approx. 15 minutes for UCF-101.
>
> Each AL cycle for UCF-101 with ~432K training frames takes an average of 50 minutes to select new annotation frames. This can be further optimized to run faster (parallel threads, efficient GPU transfer, etc) but currently we run a simple baseline version. We use 2 AL cycles (1% to 5% and 5% to 10%) to increase the annotations. We have a more detailed analysis in supplementary section 7.3 and will update that section to reflect the time usage as well.
>
> **W.2**
>
> **a.) It was not clear to me why does the model need an estimation of the uncertainty per pixel, that has to be obtained utilizing MC-Dropout. To run Eq. 1 should be highly time-consuming. Are there any other uncertainty measures that could be used? Have the authors used any alternative?**
>
> The uncertainty estimation for a frame depends on the detection for that frame, which is computed for each pixel by the model. We compute the uncertainty per pixel which is then converted to uncertainty score per frame (using non-activity suppression). We use MC-dropout based uncertainty for Eq. 1 as it is a more efficient form of uncertainty estimation compared to Bayesian NN [B1] and is easier to implement [B1, B2]. Other variations of uncertainty estimation include SpatialDropout [B3] and DropBlock [B4] that would either drop the entire feature map or drop a connected block of features instead of random dropout, with similar performance for uncertainty estimation as shown by [B2] in their experiments. These methods would still have at least the same computation time as MC-dropout since they focus on reducing training overfitting and not training time. Given time constraints we could not perform additional experiments using these methods.
>
> [B1] Yarin Gal, and Zoubin Ghahramani. "Dropout as a bayesian approximation: Representing model uncertainty in deep learning." In international conference on machine learning, 2016.
>
> [B2] Mamshad Nayeem Rizve, Kevin Duarte, Yogesh S. Rawat, and Mubarak Shah. In defense of pseudo-labeling: An uncertainty-aware pseudo-label selection framework for semi-supervised learning. In Proceedings of International Conference on Learning Representations, 2021.
>
> [B3] Jonathan Tompson, Ross Goroshin, Arjun Jain, Yann LeCun, and Christoph Bregler. Efficient object localization using convolutional networks. In Proceedings of the IEEE Conference on Computer Vision and Pattern Recognition, pp. 648–656, 2015.
>
> [B4] Golnaz Ghiasi, Tsung-Yi Lin, and Quoc V Le. Dropblock: A regularization method for convolutional networks. Advances in Neural Information Processing Systems 31, pp. 10727–10737. Curran Associates, Inc., 2018.
>
> **b.) Adaptive Proximity-aware Uncertainty (APU) objective is to select frames with temporal diversity, am I wrong? Eq. 3 presents a combination between the distance and the uncertainty with just a sum. Are both variables scaled? With a lambda of 0.5 the model assumes that the two terms are in the same range of values. An ablation study on the influence of lambda could be interesting.**
>
> Yes, APU objective is to prefer frames with more temporal diversity (via distance score) along with the uncertainty score.
>
> Both the values in Eq. 3 are normalized between 0-1 as mentioned in Line (144-145).
>
> We perform an experiment to study the influence of lambda on Eq. 3. The effect of different values of Lambda is shown in table A4 below. We observe that giving higher weight to uncertainty (Lambda=0.75) reduces overall score as it ignores proximity value. Having lower weight on uncertainty (Lambda=0.25) on the other hand promotes more distance and performs better than lambda 0.75. Although Lambda=0.25 has higher f-mAP @ 0.5, we do not optimize this hyperparameter and use equal weight in the paper (Lambda=0.5).
> | Lambda | v-mAP @ |       | f-mAP |       |
> |:------:|:-------:|:-----:|:-----:|:-----:|
> |        |   0.3   |  0.5  |  0.3  |  0.5  |
> |    0.5 |  91.12  | 73.20 | 88.72 | 71.75 |
> |   0.25 |  92.34  | 72.59 | 88.65 | 71.83 |
> |   0.75 |  90.99  | 72.09 | 88.06 | 71.67 |
>
> _Table A4: Ablation on Lambda for Eq. 3 on UCF-101 (From 5% to 10% using APU)_

---

> > ### Author Response · Authors · 2022-08-02
> > **Response to reviewer mzzJ (part 2 of 2)**
> >
> > We continue remaining response here.
> >
> > **W.2**
> >
> > **c.) Non-activity suppression: influence of Tau in the performance of the model**
> >
> > As the model is very certain on most background pixels early on in the training, we observed that this gives us a large margin between the true background pixel confidence value and the uncertain pixel confidence value. We perform ablation with different Tau values and show them in table A5 below. We observe that changing Tau affects the results slightly and hyperparameter tuning can affect the final results with small margin. We did not perform hyperparameter tuning for Tau in our proposed method.
> >
> > | Tau | v-mAP @ |       | f-mAP |       |
> > |:---:|:-------:|:-----:|:-----:|:-----:|
> > |     |   0.3   |  0.5  |  0.3  |  0.5  |
> > | 0.3 |  90.96  | 72.42 | 88.24 | 71.65 |
> > | 0.4 |  91.12  | 73.20 | 88.72 | 71.75 |
> > | 0.5 |  91.24  | 73.15 | 88.55 | 71.77 |
> >
> > _Table A5: Ablation of Tau for Non-Activity Suppression on UCF-101 (From 5% to 10% using APU)_
> >
> > **d,e) How is the proposed model integrated in the VideoCapsuleNet [9] approach? Section 3.3 needs to be extended.**
> >
> > We use the VideoCapsuleNet implementation with 1) 2D capsule routing instead of 3D routing for efficient computation, 2) dropout layers for uncertainty, 3) MGW-loss to handle sparse labels. To handle sparse labels, we get the frame-wise weight from max-Gaussian weighted method and adjust the loss using this weight. Once we have the trained model, we apply the APU algorithm using MC-dropout and distance score for each frame. We provide implementation details for the overall model in supplementary Section 7 and supplementary Figure 6. We will improve this explanation in section 3.3 and cross reference further details from supplementary section 7.1
> >
> > **Minor comments: Punctuate all equations.**
> >
> > We will address this and rectify the paper.

---

> ### Author Response · Authors · 2022-08-09
> **Review clarification**
>
> Dear reviewer mzzJ,
>
> We are sincerely thankful for the time and work you put in reviewing our paper. We hope our answers clarified your queries and if you have any more queries regarding the paper feel free to ask us any time. We will be glad to answer them.
>
> Sincerely,
>
> Authors of Paper 8487

---

### Official Review · Reviewer_2s1J · 2022-07-10

**Rating:** 3
**Confidence:** 5
**Soundness:** 2 fair
**Presentation:** 3 good
**Contribution:** 2 fair

**Summary:**

This paper presents an iterative frame-selection approach to select a subset of the most typical/useful frames from all video frames for reducing the annotation cost for the task of video action detection. In particular, the paper introduces a frame-level scoring mechanism in terms of pixel uncertainty in a video aimed at selecting the most informative frames in the video. In experiments, the proposed approach was evaluated on action detection benchmark datasets UCF-101-24 and J-HMDB-21.

**Questions:**

See above section 'Weaknesses'

**Ethics Review Area:**

["I don’t know"]

**Limitations:**

The paper seems incremental and the experiment setting seems unfair.

**Strengths And Weaknesses:**

Strengths

1. This paper works on the problem of active frame selection for the task of video action detection.

Weaknesses

1. The novelty seems incremental. The definition of frame uncertainty for selecting video frames as well as the max-gaussian weighted loss for training the action detection network is either taken from or extended from the existing approaches. The paper does not provide any new insights.

2. The paper settings seem unfair. From the settings, seems like the proposed approach only works for easy datasets with a single object on a static background but not for complex video scenarios. Working on the problem of active frame selection in a simplified/reduced constraint setting seems to be unjustified. Also, the comparison with respect to the state-of-the-art approaches seems to be unfair. For example, this paper uses the I3D encoder head with pre-trained weights from the Charades while [23] uses the I3D network trained on the Kinetics dataset.

---

> ### Author Response · Authors · 2022-08-02
> **Response to reviewer 2s1J**
>
> We sincerely thank the reviewer for the feedback and analysis on the paper. We have addressed the concerns raised by the reviewer.
>
> **W.1 Novelty is incremental. Frame uncertainty for selecting video frames and max-gaussian weighted loss is taken from or extended from existing approaches. Paper doesn’t provide any new insights.**
>
> The use of active learning for spatio-temporal video action detection is a novel task and there is no existing work focusing on this problem to the best of our knowledge. The idea of active sparse labeling is novel, there is no existing work which performs sparse labeling of videos using active learning. Existing work on active learning in the video domain selects the entire video, however, selecting important frames is very specific to the problem formulation we have proposed. Also, max-gaussian weighted loss is very specific to spatio-temporal detection, which has not been addressed from active learning perspective in the research community to the best of our knowledge. Regarding new insights, for the first time, we demonstrate that dense labeling of videos can be avoided to save annotation cost and small amounts of annotations can be sufficient to achieve performance close to fully-supervised methods. While prior works use active learning to do selection on video level (annotating all frames) or use non-active learning weakly-supervised approach, their performance was not at all comparable with fully-supervised methods.
>
> **W.2.1 The paper settings seem unfair. From the settings, seems like the proposed approach only works for easy datasets with a single object on a static background but not for complex video scenarios. Working on the problem of active frame selection in a simplified/reduced constraint setting seems to be unjustified.**
>
> There is some confusion here. UCF-101-24 videos have non-static backgrounds with complex action classes like Basketball, Cricket Bowling, Diving, Floor Gymnastics, Salsa Spin, Long Jump, Ice Dancing and more. It also has several sequences with more than one object/actor (one such instance from UCF-101 is shown in Figure 7-c). We also evaluate the YouTube-VOS dataset which has complex scenes with multiple objects.
>
> **W.2.2 Also, the comparison with respect to the state-of-the-art approaches seems to be unfair. For example, this paper uses the I3D encoder head with pre-trained weights from the Charades while [23] uses the I3D network trained on the Kinetics dataset.**
>
> Regarding the comparison with prior weakly-supervised state-of-the-art, we mention how their methods are different in Line 250-267, with most methods including [23] using complex combination of off-the-shelf object detectors such as Faster R-CNN and tube connectors on top of the I3D feature extractor. Our approach is simpler with no external off-the-shelf models. We also compare the difference with using Charades pretrained and Kinetics pretrained weight on our approach to highlight the overall effect in Table A3 below.
> |          | f-mAP @ 0.5 |       |       | v-mAP @ 0.5 |       |       |
> |----------|:-----------:|:-----:|:-----:|:-----------:|:-----:|:-----:|
> |          |      1%     |   5%  |  10%  |      1%     |   5%  |  10%  |
> | Charades |    60.7     | 66.5  | 69.3  |    59.2     | 66.4  | 69.9  |
> | Kinetics |     60.4    |  66.5 |  69.0 |     59.2    |  66.2 |  69.8 |
>
> _Table A3: Comparison between Kinetics pretrained and Charades pretrained weights for UCF-101_
>
> We will clarify further how the prior weakly-supervised approach differs in terms of annotation type (points, bounding-box, temporal information) in Table 3 in the manuscript. We hope this explanation clarifies the doubts the reviewer had regarding our approach.

---

> > ### Comment · Reviewer_2s1J · 2022-08-08
> > **We will stick to our initial ratings**
> >
> > We don't believe that the use of the existing active learning approach for the task of video action detection is novel. First, the major contributions are all taken from or extended from existing approaches. We don't see any new insights for the task of video action detection. Second, active sparse labeling has been well studied for a similar video task, semantic video segmentation,  in literature.
> >
> > The experiments indeed show that the proposed approach only improves over the existing approach in a simplified/reduced constraint setting, but not for the complex dataset, like YouTube-VOS,  which has complex scenes. On the YouTube-VOS dataset, the performance gain is marginal. It is essential to ablate the backgrounds on datasets UCF-101 and J-HMDB:  non-static vs static. The experiments seem to be unjustified without the ablation study.

---

> > > ### Author Response · Authors · 2022-08-09
> > > **Response to reviewer 2s1J comment (2 of 2)**
> > >
> > > **R. It is essential to ablate the backgrounds on datasets UCF-101 and J-HMDB: non-static vs static. The experiments seem to be unjustified without the ablation study.**
> > >
> > > Thank you for this suggestion. For a preliminary analysis, we manually separated UCF-101 classes into static/non-static based on the background motion and camera motion and shown the results at 5% and 10% annotations in table A7 below. We put classes with mostly static background and low camera motion as static classes which include [basketball, cricket bowling, diving, fencing, golf swing, horse riding, pole vault, rope climbing, salsa spin, skiing, skijet, soccer juggling, tennis swing, trampoline jump, walking with dogs] and classes with large camera motion or background movement into non-static classes which include [basketball-dunk, biking, cliff diving, skateboarding, surfing, long jump]. Some ambiguous classes [floor gymnastics, ice dancing, volleyball piking] have not been included. We observe that the action detection model performs better on static categories, which is expected, and the proposed active learning approach improves both static and non-static categories over the random baseline. We also observe that the improvement is more significant for non-static categories.
> > >
> > > |                    | 5%     |            | 10%    |            |
> > > |--------------------|--------|------------|--------|------------|
> > > |                    | Static | Non-Static | Static | Non-Static |
> > > | Random             | 76.02  | 54.12      | 76.87  | 55.94      |
> > > | Our                | 77.01  | 59.93      | 79.44  | 61.35      |
> > >
> > > _Table A7: Preliminary score for static vs non-static samples. Shown v-mAP @ 0.5 mIoU at 5% and 10% annotation for UCF-101._
> > >
> > > We will perform this analysis on static vs non-static evaluation in more depth. Please note that we might not be able to get these scores before the deadline as we only have one more day, but we will try our best. We will include this analysis in the final version.

---

> > > ### Author Response · Authors · 2022-08-09
> > > **Response to reviewer 2s1J comment (1 of 2)**
> > >
> > > We thank the reviewer for their time and providing us another opportunity to address the raised concerns. We are sorry that the previous response did not answer the raised questions adequately. Here we will try our best to answer all the questions.
> > >
> > > **R. We don't believe that the use of the existing active learning approach for the task of video action detection is novel.**
> > >
> > > We agree with the reviewer that just use of the existing active learning approach for the task of video action detection is not novel. The proposed work is *not* just using an existing active learning approach for the task of video action detection. We propose a novel approach to it and there are novel components (such as APU scoring mechanism and MGW-Loss) which are proposed specifically for video action detection. Direct use of existing active learning approaches does not work well for video action detection (as shown in the paper). Also, the use of active learning for video action detection task has not been studied before to the best of our knowledge.
> > >
> > > **R. First, the major contributions are all taken from or extended from existing approaches**
> > >
> > > We take this criticism positively, but it is not clear to the authors which existing approaches the reviewer is referring to. We are not aware of any existing works on video action detection which use active learning.
> > >
> > > **R. We don't see any new insights for the task of video action detection**
> > >
> > > Regarding new insights, for the first time, we demonstrate that dense labeling of videos can be avoided for video action detection to save annotation cost and small amounts of annotations can be sufficient to achieve performance close to fully-supervised methods. While prior works use active learning to do selection on video level (annotating all frames) or use non-active learning weakly-supervised approach, their performance was not at all comparable with fully-supervised methods. We also demonstrate why existing active learning approaches are not sufficient for video action detection.
> > >
> > > **R. Second, active sparse labeling has been well studied for a similar video task, semantic video segmentation**
> > >
> > > In this work, our focus is on video action detection. We have shown some results on the task of video segmentation to merely demonstrate the generalization capability of the proposed method. Also, regarding the reviewers claim *‘has been well studied for semantic video segmentation’*, sorry, but we did not find any recent works on semantic video segmentation based on active learning (the only one we found from 2011 is already discussed in the related work). It will be great if the reviewer can share the recent referred papers, discussing them in related work will further strengthen our work.
> > >
> > > **R. The experiments indeed show that the proposed approach only improves over the existing approach in a simplified/reduced constraint setting, but not for the complex dataset, like YouTube-VOS, which has complex scenes. On the YouTube-VOS dataset, the performance gain is marginal.**
> > >
> > > Sorry, but there is some confusion here. It is not clear why the reviewer considers this as a simplified/reduced constraint setting. Both the datasets used for the experiments are well known benchmark datasets for video action detection and we are using them *as it is* without any simplifications or constraint settings. Also, for YouTube-VOS, in fact the performance gain over baseline and existing approaches is even more (~18%-23% improvement on overall score) when compared against UCF-101 and JHMDB datasets.

---

### Official Review · Reviewer_cBj8 · 2022-07-11

**Rating:** 6
**Confidence:** 4
**Soundness:** 4 excellent
**Presentation:** 4 excellent
**Contribution:** 4 excellent

**Summary:**

The paper proposes an efficient learning paradigm for the task of video action detection where all the frames in a video need not be labelled with spatio-temporal annotations. The task is important since the cost of annotations increases as the task granularity increases from action classification to action detection. The paper uses an active learning scheme to select most informative frames to annotate, and manages to achieve the performance comparable to fully-annotated approaches while using a fraction of the labelled frames


**Questions:**

See "Weaknesses" above

**Limitations:**

Yes

**Strengths And Weaknesses:**

## Strengths
- The task itself is important and the conclusions are worth looking into for any action detection approaches
- The approach is simple, intuitive and most likely general enough to be deployed in the setup of fine-grained video tasks

## Weaknesses
- Dataset and baseline in Table 1 and 2
    - The random baseline method achieves f-mAP@0.5 score of 69.3 with 10% data as compared to the proposed approach’s score of 71.7
    - Given the claim that the proposed approach reduces the need for annotation by 90%, the baseline itself seems to be not that far away, do the authors have an estimate of how much annotation does the baseline method need to achieve the performance of fully-supervised approaches?
    - If the answer is, say less than 50%, does that say anything about the dataset itself? Do the authors think that experimenting with more complex dataset might help bring out the efficacy of their approach?
- Non-activity suppression and relative area of spatio-temporal predictions (para on line 154)
    - The spatio-temporal annotations should have different relative area that they occupy in a frame, i.e, they lie on a spectrum same as MS-COCO object bounding boxes
    - Since the approach needs to contend with uncertainty being influenced by background, do the authors have some breakdown on performance w.r.t. relative area of spatio-temporal annotations? This is similar to AP_S, AP_M and AP_L evaluation procedure in MS-COCO object detection task
    - The reason for asking this is to check whether the algorithmic choice in the paper induces some preference over the relative area of spatio-temporal predictions
- Annotation cost of far-away frames v/s nearby frames in a video?
    - The analysis made in the paper assumes a uniform frame annotation cost regardless of where the frame is located w.r.t. annotated frames
    - My guess is that, in real life, annotating chunks or blocks of frames is easier than annotating frames one-by-one far away from each other in a video. This suggests that the cost of annotating frames will be lower near the annotated frames, or the selection process can actually select a block of frames at a given location without sacrificing too much of the cost
    - This argument partially contradicts the design choices made in the paper about not choosing proximal frames using Gaussian kernels
    - Do the authors have any comment on the above? Did the authors perform real-life analysis of annotation cost on the UCF and JHMDB datasets and validate their assumption that annotation cost of a frame is same regardless of where it occurs in a partially annotated video?

### Low priority
- Use of pre-trained weights on Charades dataset (line 215)
    - Do the authors have any intuition regarding the selection of Charades dataset? Is the approach expected to perform differently with different pre-training datasets such as Kinetics?
- Scalability of the approach to larger datasets
    - Did the authors experiment with larger datasets to check whether the approach generalizes with scale of the datasets?

### Nice to haves:
- It would be nice to have the word “utilize” substituted with “use” throughout the text. It will help reduce cognitive overload.
- The supplementary has a lot of sections, but are not cross-referenced in the main paper. It would be nice to have that so the reader can go into details on a particular section if they want to

---

> ### Author Response · Authors · 2022-08-02
> **Response to reviewer cBj8 (part 1 of 2)**
>
> We sincerely thank the reviewer for the valuable feedback and analysis on the paper. We have addressed the questions and concerns raised by the reviewer.
>
> **W.1.1 Do the authors have an estimate of how much annotation does the baseline method need to achieve the performance of fully-supervised approaches?**
>
> We extend the baselines as well as our method further in supplementary Figure 4 where we see that the baselines eventually converge close to fully-supervised scores with additional annotations. Our frame selection approach gets higher score earlier and saturates (20% our = v-mAP=73.8, f-mAP=73.0 @ 0.5) and (40% our = v-mAP=74.6, f-mAP=73.5) vs 100% v-mAP=75.1, f-mAP=74.0 in UCF-101 , demonstrating need for less annotation to achieve higher scores.
> We also extend the baseline random method and compare with our proposed method in table A6 below. We will update the manuscript to highlight this table.
> |  | Random |  | OUR | |
> |:---:|:---:|:---:|:---:|:---:|
> |  | v-mAP @ 0.5 | f-mAP @ 0.5 | v-mAP @ 0.5 | f-mAP @ 0.5 |
> | 10%  | 69.9 | 69.3 | 73.2 | 71.7 |
> | 20%  | 69.2 | 69.2 | 73.8 | 73.0 |
> | 30%  | 72.7 | 72.3 | 74.3 | 73.3 |
> | 40%  | 73.1 | 73.2 | 74.6 | 73.5 |
> | **100% ** | **75.1** | **74.0** | **75.1** | **74.0** |
>
> _Table A6: Comparison for random baseline with proposed method for extended annotations_
>
> **W.1.2 Do the authors think that experimenting with more complex dataset might help bring out the efficacy of their approach?**
>
> As we get close to fully-supervised scores with 40% of the dataset, it shows the dataset is not extremely complex. However, annotating 40% of UCF-101 for dense spatio-temporal detection still gets costly and prior weakly-supervised methods fail to perform close to fully-supervised regardless of the dataset complexity. As suggested by the reviewer, we have analyzed our approach on a more complex Youtube-VOS dataset for a different task in table 4, where the overall score at 30% annotation for baseline (42.5) is much lower than the proposed approach (66.7), which better demonstrates the efficacy of the proposed method.
>
> **W.2 Breakdown on performance w.r.t. relative area of spatio-temporal annotations?**
>
> Similar to MS-COCO dataset, we also separated the evaluation set into small, medium and large based on each video’s average activity area with respect to the frame size [Small < 702, Medium >= 702 and < 1302, Large >= 1302 in square pixels for UCF-101]. With this distinction, we had 267 small videos, 479 medium videos and 164 large videos. As all video classes don’t have all three variations, some of the classes don’t exist in small and large sets. As such, we only compare the common classes across all three sets for equivalent comparison. From table A2, we see that our approach selects samples to improve small and medium sized actions more compared to random selection baseline. We will add this analysis in the manuscript.
> |        |     Our (APU)   ||     Random     ||
> |--------|:---------:|:-----:|:------:|:-----:|
> |        | v-mAP     | f-mAP | v-mAP  | f-mAP |
> | Small  | 58.45     | 60.13 | 52.78  | 52.94 |
> | Medium | 84.26     | 81.33 | 81.73  | 79.13 |
> | Large  | 90.20     | 88.34 | 88.96  | 86.44 |
>
> _Table A2: Size to performance comparison for UCF-101 at 10% annotation. Scores shown for v-mAP, f-mAP @ 0.5 IoU for common classes in all three sets._
>
> **W.3 Annotation cost of far-away frames v/s nearby frames in a video.**
>
> This is an interesting aspect which could give real-life analysis of annotating videos. However, the annotation cost of a frame can be very challenging to estimate due to high variability in videos (static vs high motion, dynamics, scene complexity, etc.). On one hand annotation effort might be less due to similar surroundings in neighboring frames, while it might also have a lot of redundant information making the annotation less useful. The videos with slow motion can have more accurate pseudo-labels for nearby frames, so it would be redundant to annotate them and increase overall cost. However, this might be different in videos with fast motion, in which case uncertainty can guide in selecting better frames along with distance factor. While we do not have such real-life analysis on annotation cost based on frequency of annotation, this would be a good future direction for hybrid video-frame annotation methods.

---

> > ### Author Response · Authors · 2022-08-02
> > **Response to reviewer cBj8 (part 2 of 2)**
> >
> > We continue the remaining response here.
> >
> > **LP.1 Use of pre-trained weights on Charades and Kinetics.**
> >
> > We don’t have any bias to use Charades pretrained weights beyond the general convention for recent video understanding tasks. We evaluated the performance of the models using pre-trained Kinetics weight and found that it is similar in relative performance between the active learning cycles. We show the difference between these two pretrained weights in table A3 below.
> >
> > |          | f-mAP @ 0.5 |       |       | v-mAP @ 0.5 |       |       |
> > |----------|:-----------:|:-----:|:-----:|:-----------:|:-----:|:-----:|
> > |          |      1%     |   5%  |  10%  |      1%     |   5%  |  10%  |
> > | Charades |    60.7     | 66.5  | 69.3  |    59.2     | 66.4  | 69.9  |
> > | Kinetics |     60.4    |  66.5 |  69.0 |     59.2    |  66.2 |  69.8 |
> >
> > _Table A3: Comparison between Kinetics pretrained and Charades pretrained weights for UCF-101_
> >
> > **LP.2 Scalability of the approach to larger datasets**
> >
> > We agree with the reviewer that experimenting with larger datasets will better demonstrate the generalizability of the proposed approach with scale of datasets. However, we were not able to perform such an experiment due to lack of larger datasets.
> >
> > **Nice to haves:** We will amend the paper to substitute the word ‘utilize’ and cross-reference the sections from supplementary in the main paper to add extra details and improve readability for everyone.

---

> ### Author Response · Authors · 2022-08-09
> **Review clarification**
>
> Dear Reviewer cBj8,
>
> We are sincerely thankful for the time and work you put in reviewing our paper. We hope our answers clarified your queries and if you have any more queries regarding the paper feel free to ask us any time. We will be glad to answer them.
>
> Sincerely,
>
> Authors of Paper 8487

---

### Official Review · Reviewer_uPyB · 2022-07-11

**Rating:** 6
**Confidence:** 4
**Soundness:** 3 good
**Presentation:** 4 excellent
**Contribution:** 4 excellent

**Summary:**

This paper proposes an approach for automatically selecting a few frames from training videos to be annotated by humans and for using these annotations to train models for spatio-temporal video understanding tasks, such as spatio-temporal action detection and video object segmentation.

1. In contrast to prior work on active learning strategies for video understanding, which choose which videos to fully-annotate, the proposed approach chooses specific frames under a specific annotation budget.
2. The first contribution is an approach for active sparse labeling in videos, which utilizes frame-level uncertainty of the model to identify frames that need to be annotated. Motivated by the task of action detection, it also ensures that a) a set of temporally diverse frames are selected (instead of consecutive frames with redundant information), b) that background pixels don’t influence the selection (since the model might be fairly certain for them).
3. The second contribution is a training regime from learning from sparsely-labeled frames, which involves interpolating annotations in the rest of frames, and a loss (max-gaussian weighted loss) that discounts the penalty from wrong predictions at frames that are distant from the ones that have human annotations.
4. The proposed method is evaluated on three datasets: UCF-101 and J-HMDB for spatio-temporal action detection, and YouTube-VOS for video object segmentation, where it is shown that it outperforms other baselines for frame selection given a fixed annotation budget.


**Questions:**

Questions for rebuttal
1. How were the videos selected in Figure 6 (c-d)? Comparing with active learning methods for video selection would strengthen the paper.  It would also be feasible to try hybrid approaches, e.g. choose videos samples with active learning and then use equidistant frames within them, to fit more diverse frames within a given annotation budget.
2. Additional graphs: It would be nice to extend some graphs to more than 9-10% of annotated frames. It would also be interesting to see the performance over the different steps of active learning, as more frames are added.

Suggestions
1. What is the impact of \sigma for the MGW-Loss?
2. How many active learning cycles are used?
3. What happens if you start from the same percentage of annotated frames in UCF-101/JHMDB?



**Limitations:**

Yes

**Strengths And Weaknesses:**

Strengths
========

1. The paper addresses an important challenge in the action detection literature, namely how to train systems with fewer labeled data. This is especially important for tasks that require fine-grained annotations, such as bounding boxes or segmentation masks, which is the focus of this paper. Focusing on spatio-temporal video understanding tasks and on selecting frames for annotation instead of whole videos clearly distinguishes this work from prior approaches. The review of prior approaches also seems adequate.
2. The proposed approach is simple, but well-motivated (both intuitively and based on ablations). It builds upon MC-dropout for uncertainty-based active learning, but also takes into account the nature of videos with the proposed Adaptive Proximity-aware Uncertainty, which outperforms other uncertainty-based selection methods (Fig. 4).  The proposed loss also outperforms training simply with interpolated annotations or only on annotated frames (Fig. 5).
3. The proposed approach outperforms other frame selection baselines (such as choosing equidistant frames or using approaches that were proposed for images).
4. The paper contains multiple qualitative examples, which clearly demonstrate the issues of related baselines.
5. The proposed approach does not require an actor/object detector.

Weaknesses
===========
1. Since this seems to be the first approach for active learning for these particular video understanding tasks (spatio-temporal action detection, video object detection), choosing the right baselines (in Table 1) is crucial. The evaluation could be strengthened by comparing with video selection methods, such as approaches from “What do I Annotate Next? An Empirical Study of Active Learning for Action Localization”[60].
2. Method is only evaluated for selecting frames up to 9-10% frame annotation percentage: It would make sense to investigate how many annotated frames are needed to reach the fully-supervised performance (if possible).
3. Comparison with state-of-the-art: Table 3 is comparing methods with very different levels of supervision and architectures. Although it is nice that the proposed method leads to the best performance, it should be clear in the table that the comparison is not apples to apples. It would be helpful to add more details about the type of annotations (points?, temporal bounds? bounding boxes?) and the performance of each method given full-supervision if available (this will help clarify whether the models trained were much weaker than the capsule-based network). For completeness, the state-of-the-art fully-supervised method for each dataset should also be included.
4. More details about the training loss (margin-loss, binary-cross entropy loss) and the metrics (do they refer to temporal IoUs, spatio-temporal IoUs etc) are needed.

---

> ### Author Response · Authors · 2022-08-02
> **Response to reviewer uPyB (part 1 of 2)**
>
> We sincerely thank the reviewer for the valuable feedback and analysis on the paper. We have addressed the questions and concerns raised by the reviewer.
>
> **W.1, Q1.2  Comparing with video selection method such as [60] in Table 1.**
>
> We agree that comparing with video selection methods in Table 1 will strengthen the evaluation.  Currently we compare with uncertainty sampling (US) for video level selection (Figure 6(c-d), Table 8 (supplementary), and Table 9 (supplementary)). In our preliminary experiments we observed that uncertainty scores at video-level (as proposed by [60]) performs similar to random selection for this task. Therefore, we use a stronger baseline where uncertainty is computed at pixel-level which is averaged over all the pixels (Equation 1) and then all the frames to determine video-level uncertainty. This baseline is used for Figure 6(c-d) and Table 8&9 in supplementary. We will include these baseline scores in Table 1 and clarify this baseline approach. The other methods from [60] (LAL, MCLE) use additional data sources and their source code is not publicly available, therefore we could not compare with these variations.
>
> **W.2, Q.2  Additional graphs beyond 10% annotation.**
>
> The extended analysis beyond 10% annotations is provided in the supplementary material (Figure 4 and section 5.5) for UCF-101 and J-HMDB datasets. We observe that as we add more frames, the temporal density of annotation increases and all the methods start to converge towards the fully-supervised baseline with very little variation. The proposed method saturates earlier as compared to other baselines, demonstrating that the proposed frame selection can achieve better performance at a lower cost. We perform some more experiments and the results for extended analysis for the proposed method are shown in table A1 below. We will add these details in the main paper and supplementary details with cross-reference.
>
> |  | UCF-101 |  |  | J-HMDB |  |
> |---|:---:|:---:|---|:---:|:---:|
> |  | v-mAP @ 0.5 | f-mAP @ 0.5 |  | v-mAP @ 0.5 | f-mAP @ 0.5 |
> | 10%  | 73.2 | 71.7 | 9%  | 74.0 | 74.5 |
> | 20%  | 73.8 | 73.0 | 18% | 74.9 | 74.5 |
> | 30%  | 74.3 | 73.3 | 27% | 75.2 | 74.6 |
> | 40%  | 74.6 | 73.5 | 35% | 75.3 | 74.6 |
> | **100% ** | **75.1** | **74.0** | **100%** | **75.7** | **74.9** |
>
> _Table A1: Results for proposed method with extended annotations_
>
> **W.3. Comparison with state-of-the-art in Table 3**
>
> We agree with the reviewer that the comparisons provided in Table 3 are not apples to apples,  and we compared these methods to have a comprehensive evaluation. We agree having more information on annotation type (point, bounding boxes, temporal information) in the table will clarify the comparison and we will add these details in the table. We will also add the fully-supervised results of available methods. We do provide an explanation of these methods from Table 3 in the text (Line 250-267), on which we will add more details to highlight the differences in the annotations. We provide results of prior fully-supervised methods in supplementary Table 4 for a more complete picture as well. We will cross-reference those in the main paper clearly.
>
> **W.4. More details about the training loss and metrics**
>
> Following prior action detection works ([22][77]), we compute the spatial IoU for each frame per class to get the f-mAP score and compute the spatio-temporal IoU per video per class to get the v-mAP score. Training uses binary cross-entropy loss as the detection loss and margin loss as classification loss. We will further expand on the metrics in section 4 and include the details of these loss functions in section 3.3.
>
> **Q1.1 How were videos selected in figure 6(c-d)?**
>
> We compute pixel level uncertainty which is averaged over all the pixels in a frame (Equation 1) and then averaged over all frames in a video to get the video level score. This detail was missed earlier and we will add this to section 4.4 (line 300-306).
>
> **Q1.3 It would be feasible to try hybrid approaches to fit more diverse frames within a given annotation budget.**
>
> This is indeed an interesting future direction for the proposed work to reduce annotation cost while reducing the performance gap with fully-annotated works (Line 314). We perform a preliminary experiment which uses video sampling (uncertainty-based) with equidistant frame selection. We selected a total of 20% videos with equidistant frames annotated (an overall 1% frame annotation) and obtain v-mAP= 53.3 and f-mAP=55.0 @ 0.5 IoU for UCF-101 which is lower than 1% equidistant annotation from all videos in Table 1 (v-mAP=61.7, f-mAP=61.8 @ 0.5 IoU). We will add this approach as a baseline in Table 1.

---

> > ### Author Response · Authors · 2022-08-02
> > **Response to reviewer uPyB (part 2 of 2)**
> >
> > We continue the remainder of our response here
> >
> > **S.1. Impact of \sigma for MGW-Loss**
> >
> > We analyze the impact of \sigma for MGW-loss in more detail in supplementary section 3.1 and supplementary Table 3. Sigma controls the weight given to pseudo-label based on their proximity to the real labels, so we evaluate the effect of high to low sigma in a controlled setting. Compared to the sigma used in this work (sigma=1.3 with v-mAP=73.2), low sigma gives little weight to pseudo-labels and does not use them fully (v-mAP=69.9), and high sigma gives too much weight to pseudo-labels and affects learning (v-mAP=71.5). The weight distribution for different sigma values are also shown in supplementary Figure 2.
> >
> > **S.2. How many active learning cycles are used?**
> >
> > We use a total of 2 active learning cycles each for UCF-101 and J-HMDB. Each increase in annotation takes 1 active learning cycle. We also compare this with smaller increments where more active learning cycles are required. This is provided in supplementary Table 5 to show the analysis on the effect of the number of active learning cycles on the performance.
> >
> > **S.3. What happens if you start from the same percentage of annotated frames in UCF-101/JHMDB?**
> >
> > First we clarify on why J-HMDB has a 3% start rate. J-HMDB is a smaller dataset with only ~22K total frames annotated. The training set has 660 videos, so to keep the same setting as UCF-101 we select 1 frame annotation per video which gives us 3% of total annotation. If we limit to 1% for J-HMDB, we have to exclude 2/3 of those videos, giving us only 220 initial frames. This makes initial model training harder [v-mAP=53.6, f-mAP=52.9 @ 0.5 IoU].

---

> ### Author Response · Authors · 2022-08-09
> **Review clarification**
>
> Dear Reviewer uPyB,
>
> We are sincerely thankful for the time and work you put in reviewing our paper. We hope our answers clarified your queries and if you have any more queries regarding the paper feel free to ask us any time. We will be glad to answer them.
>
> Sincerely,
>
> Authors of Paper 8487

---

### Meta-Review · Area_Chair_WqSh · 2022-08-28

**Recommendation:** Accept
**Confidence:** Certain

**Metareview:**

Paper was reviewed by four reviewers and received: 2 x Weak Accepts, 1 x Reject and 1 x Accept. Reviewers argued that the task is interesting and approach is simple and well motivated. Some raised concerns included: (1) lack of additional comparisons and baselines, (2) lack of discussion regarding far-away frames v/s nearby frames for annotation, (3) lack of novelty and (4) fairness of evaluation. Three out of four reviewers were reasonably convinced by the rebuttal and argue for acceptance. [2s1J] remains concerned about (3) and (4). This was carefully considered by AC. Because no specific papers were provided by [2s1J] to support the claims of lacking novelty, and given the remaining positives reviews, AC is inclined to accept the paper.

**Award:**

No

---

### Decision · Program_Chairs · 2022-09-14

Accept